# Rab12 is a regulator of LRRK2 and its activation by damaged lysosomes

Xiang Wang[1†], Vitaliy V Bondar[1†‡], Oliver B Davis[1], Michael T Maloney[1], Maayan Agam[1], Marcus Y Chin[1], Audrey Cheuk-Nga Ho[1§], Rajarshi Ghosh[1], Dara E Leto[1], David Joy[1], Meredith EK Calvert[1], Joseph W Lewcock[1], Gilbert Di Paolo[1], Robert G Thorne[1,2], Zachary K Sweeney[1#], Anastasia G Henry[1]*

[1]Denali Therapeutics, South San Francisco, United States; [2]Department of Pharmaceutics, University of Minnesota, Minneapolis, United States

**Abstract** Leucine-rich repeat kinase 2 (LRRK2) variants associated with Parkinson's disease (PD) and Crohn's disease lead to increased phosphorylation of its Rab substrates. While it has been recently shown that perturbations in cellular homeostasis including lysosomal damage can increase LRRK2 activity and localization to lysosomes, the molecular mechanisms by which LRRK2 activity is regulated have remained poorly defined. We performed a targeted siRNA screen to identify regulators of LRRK2 activity and identified Rab12 as a novel modulator of LRRK2-dependent phosphorylation of one of its substrates, Rab10. Using a combination of imaging and immunopurification methods to isolate lysosomes, we demonstrated that Rab12 is actively recruited to damaged lysosomes and leads to a local and LRRK2-dependent increase in Rab10 phosphorylation. PD-linked variants, including LRRK2 R1441G and VPS35 D620N, lead to increased recruitment of LRRK2 to the lysosome and a local elevation in lysosomal levels of pT73 Rab10. Together, these data suggest a conserved mechanism by which Rab12, in response to damage or expression of PD-associated variants, facilitates the recruitment of LRRK2 and phosphorylation of its Rab substrate(s) at the lysosome.

*For correspondence:
henry@dnli.com

†These authors contributed equally to this work

Present address: ‡REGENXBIO Inc, Rockville, United States; §Cellares, South San Francisco, United States; #Interline Therapeutics Inc, Brisbane, United States

## Editor's evaluation

This valuable study shows that Rab12 regulates LRRK2 activation via damaged lysosomes. The strength of evidence supporting the claim is compelling. Although key questions about the mechanism remain unanswered, these findings provide a useful template for further research.

## Introduction

Coding variants in *LRRK2* can cause monogenic Parkinson's disease (PD), and coding and non-coding variants in *LRRK2* are associated with increased risk for developing sporadic PD and Crohn's disease (*Blauwendraat et al., 2020*; *Hui et al., 2018*; *Kluss et al., 2019*). The majority of pathogenic LRRK2 variants cluster within its Roc-COR (Ras of complex proteins/C-terminal of Roc) GTPase tandem domain or kinase domain and contribute to disease risk by ultimately increasing LRRK2's kinase activity (*Alessi and Sammler, 2018*; *Kalogeropulou et al., 2022*; *Sheng et al., 2012*). LRRK2 phosphorylates a subset of Rab GTPases, including Rab10 and Rab12. Excessive Rab phosphorylation can perturb interactions between Rabs and downstream effectors, which impairs various aspects of membrane trafficking including lysosomal function (*Pfeffer, 2018*; *Steger et al., 2017*; *Steger et al., 2016*). LRRK2 localizes primarily to the cytosol in an inactive state, and higher-order oligomerization and membrane recruitment appear to be required for LRRK2 activation and Rab phosphorylation (*Berger et al., 2010*; *Biskup et al., 2006*; *Greggio et al., 2008*; *Schapansky*

**eLife digest** Lysosomes are cellular compartments tasked with breaking down large molecules such as lipids or proteins. They perform an essential role in helping cells dispose of obsolete or harmful components; in fact, defects in lysosome function are associated with a range of health conditions. For instance, many genes associated with an increased risk of developing Parkinson's disease code for proteins required for lysosomes to work properly, such as the kinase LRRK2.

Previous work has shown that this enzyme gets recruited to the surface of damaged lysosomes, where it can modulate the function of another set of molecular actors by modifying them through a chemical process known as phosphorylation. Such activity is increased in harmful versions of LRRK2 linked to Parkinson's disease. However, the molecular mechanisms which control LRRK2 activity or its recruitment to lysosomes remain unclear.

To examine this question, Wang, Bondar et al. first performed a targeted screen to identify proteins that can regulate LRRK2 activity. This revealed that Rab12, one of molecular actors that LRRK2 phosphorylates, can in turn modulate the activity of the enzyme. Further imaging and biochemical experiments then showed that Rab12 is recruited to damaged lysosomes and that this step was in fact necessary for LRRK2 to also relocate to these compartments. The data suggest that this Rab12-driven recruitment process increases the local concentration of LRRK2 near its Rab targets on the membrane of damaged lysosomes, and therefore leads to enhanced LRRK2 activity. Crucially, Wang, Bondar et al. showed that Rab12 also plays a role in the increased LRRK2 activity observed with two Parkinson's disease-linked mutations (one in LRRK2 itself and one in another lysosomal regulator, VPS35), suggesting that increased LRRK2 concentration on lysosomes may be a conserved mechanism that leads to increased LRRK2 activity in disease.

Overall, these results highlight a new, Rab12-dependent mechanism that results in enhanced activity at the lysosomal membrane with variants associated with Parkinson's disease, and for LRRK2 in general when lysosomes are damaged. This knowledge will be helpful to develop therapeutic strategies that target LRRK2, and to better understand how increased LRRK2 activity and lysosomal injury may be linked to Parkinson's disease.

*et al., 2014*). Recent work suggests that interactions between LRRK2 and its phosphorylated Rab substrates help maintain LRRK2 on membranes and result in a cooperative, feed-forward mechanism to promote additional Rab phosphorylation (*Vides et al., 2022*). However, it is not clear what mechanisms promote the initial recruitment of LRRK2 to membranes to trigger Rab phosphorylation or whether increased LRRK2 membrane association is a common driver of the elevated LRRK2 activity observed in PD.

Endolysosomal genes that modify PD risk and lysosomal damage can also increase LRRK2 activation and phosphorylation of its Rab substrates (*Bonet-Ponce et al., 2020*; *Eguchi et al., 2018*; *Herbst et al., 2020*; *Liu et al., 2018*; *Mir et al., 2018*; *Purlyte et al., 2018*). Rab29 and the retromer subunit VPS35, proteins that are genetically associated with PD and play key roles in lysosomal function by regulating sorting between the endolysosomal system and the trans-Golgi network, can modulate LRRK2 activity, as overexpression of Rab29 or expression of the pathogenic VPS35 D620N variant lead to significantly elevated LRRK2-mediated phosphorylation of Rab10 and other Rab substrates (*Liu et al., 2018*; *Mir et al., 2018*; *Purlyte et al., 2018*). LRRK2 kinase activity also appears to be increased in nonhereditary idiopathic PD patients (*Di Maio et al., 2018*; *Fraser et al., 2016*; *Wang et al., 2021*), and emerging data suggest that lysosomal damage more broadly may be a common trigger for LRRK2 activation. Lysosomotropic agents that disrupt the endolysosomal pH gradient or puncture lysosomal membranes enhance LRRK2 recruitment to damaged lysosomes and result in increased Rab10 phosphorylation (*Bonet-Ponce et al., 2020*; *Eguchi et al., 2018*; *Herbst et al., 2020*; *Wang et al., 2021*). Several hypotheses around the purpose of LRRK2 recruitment to damaged lysosomes have been proposed, including promotion of lysosomal membrane repair or clearance of lysosomal content through exocytosis or sorting of vesicles away from damaged lysosomes (*Bonet-Ponce et al., 2020*; *Eguchi et al., 2018*; *Herbst et al., 2020*). There is a clear need to better define how LRRK2 is recruited to lysosomes upon damage, how these steps translate to LRRK2 activation, and how broadly conserved these mechanisms are in PD.

Here, we identify Rab12 as a novel regulator of LRRK2 activity. Rab12 regulates LRRK2-dependent phosphorylation of Rab10 and mediates LRRK2 activation in response to lysosomal damage. We demonstrate that Rab12 promotes Rab phosphorylation by recruiting LRRK2 to lysosomes following lysosomal membrane rupture. Pathogenic PD variants including VPS35 D620N and LRRK2 R1441G also result in increased levels of LRRK2 and pT73 Rab10 on lysosomes. Together, our data delineate a conserved mechanism by which LRRK2 activity is regulated basally and in response to lysosomal damage and genetic variants associated with disease.

## Results

### siRNA-based screen identifies Rab12 as a key regulator of LRRK2 kinase activity

Although a subset of 14 Rab GTPases has been clearly established as LRRK2 substrates, increasing data suggest a reciprocal relationship exists in which Rab proteins may also contribute to LRRK2 membrane association and activation (*Vides et al., 2022*; *Liu et al., 2018*; *Purlyte et al., 2018*; *Gomez et al., 2019*). Previous studies have shown that overexpression of one such LRRK2 substrate, Rab29, can increase LRRK2-dependent phosphorylation of Rab10 by promoting its membrane association at the Golgi complex (*Liu et al., 2018*; *Purlyte et al., 2018*; *Gomez et al., 2019*). However, additional work in RAB29 KO models demonstrated that LRRK2 activity was minimally impacted by loss of Rab29, suggesting Rab29 does not regulate LRRK2 activity under physiological conditions (*Kalogeropoulou et al., 2020*). To determine whether any LRRK2-Rab substrates were required for LRRK2 kinase activity, we performed a targeted siRNA screen on 14 Rab genes in human A549 cells that endogenously express both LRRK2 and Rab10. Rab10 phosphorylation was chosen as the endpoint to assess LRRK2 activation as it is an established readout of LRRK2 kinase activity that has been routinely used in preclinical and clinical settings (*Wang et al., 2021*; *Fan et al., 2018*; *Jennings et al., 2022*). The levels of Rab10 phosphorylation were measured using a previously described quantitative Meso Scale Discovery (MSD)-based assay (*Wang et al., 2021*). Greater than 50% knockdown of gene expression of each target was demonstrated using RT-qPCR-based analysis, and we confirmed that knockdown of the positive controls LRRK2 and RAB10 attenuated the phospho-Rab10 signal as expected (*Figure 1A and B* and *Figure 1—figure supplement 1*). RAB12 was the only hit gene whose knockdown significantly reduced Rab10 phosphorylation (*Figure 1A*). We confirmed that RAB12 knockdown reduced gene expression and led to a reduction in Rab12 protein levels (*Figure 1C–E*). RAB12 knockdown did not impact the levels of LRRK2 or Rab10, suggesting that Rab12 mediates Rab10 phosphorylation by regulating LRRK2's activity rather than the stability of LRRK2 or Rab10 (*Figure 1B, D, and F* and *Figure 1—figure supplement 1*). We also confirmed previous observations that Rab29 does not regulate LRRK2 activity under endogenous expression conditions as RAB29 knockdown or genetic deletion did not impact Rab10 phosphorylation in A549 cells (*Figure 1A* and *Figure 1—figure supplement 1*). Together, these data identify Rab12 as a novel regulator of LRRK2 kinase activity.

### Rab12 deletion attenuates LRRK2-dependent phosphorylation of Rab10

To confirm that Rab12 regulates LRRK2-dependent Rab10 phosphorylation, we performed MSD and western blot analysis in RAB12 KO A549 cells. We demonstrated that loss of Rab12 significantly impairs Rab10 phosphorylation at T73, showing a comparable reduction to that observed with loss of LRRK2 (*Figure 2A and B* and *Figure 2—figure supplement 1*). Total Rab10 levels were not reduced with RAB12 deletion and, in fact, were elevated in two out of three RAB12 KO clones assessed, confirming that the impact of loss of Rab12 on Rab10 phosphorylation cannot be explained by an effect on the protein levels of Rab10 (*Figure 2C*). Rab12 is itself a substrate for LRRK2, and we next explored whether LRRK2-mediated phosphorylation of Rab12 contributed to LRRK2 activation. To assess this, we generated doxycycline-inducible stable cell lines in the RAB12 KO cell background to allow overexpression of wildtype (WT) or a phospho-deficient mutant Rab12 in which the LRRK2 phosphorylation site (S106) was converted to an alanine (*Figure 2D*). Overexpression of either WT Rab12 or Rab12 S106A restored Rab10 phosphorylation at T73 and did not impact LRRK2 levels (*Figure 2E and F*). This finding was further confirmed using Rab12 S106A KI cells generated using CRISPR-Cas9 (*Figure 2—figure supplement 1*).

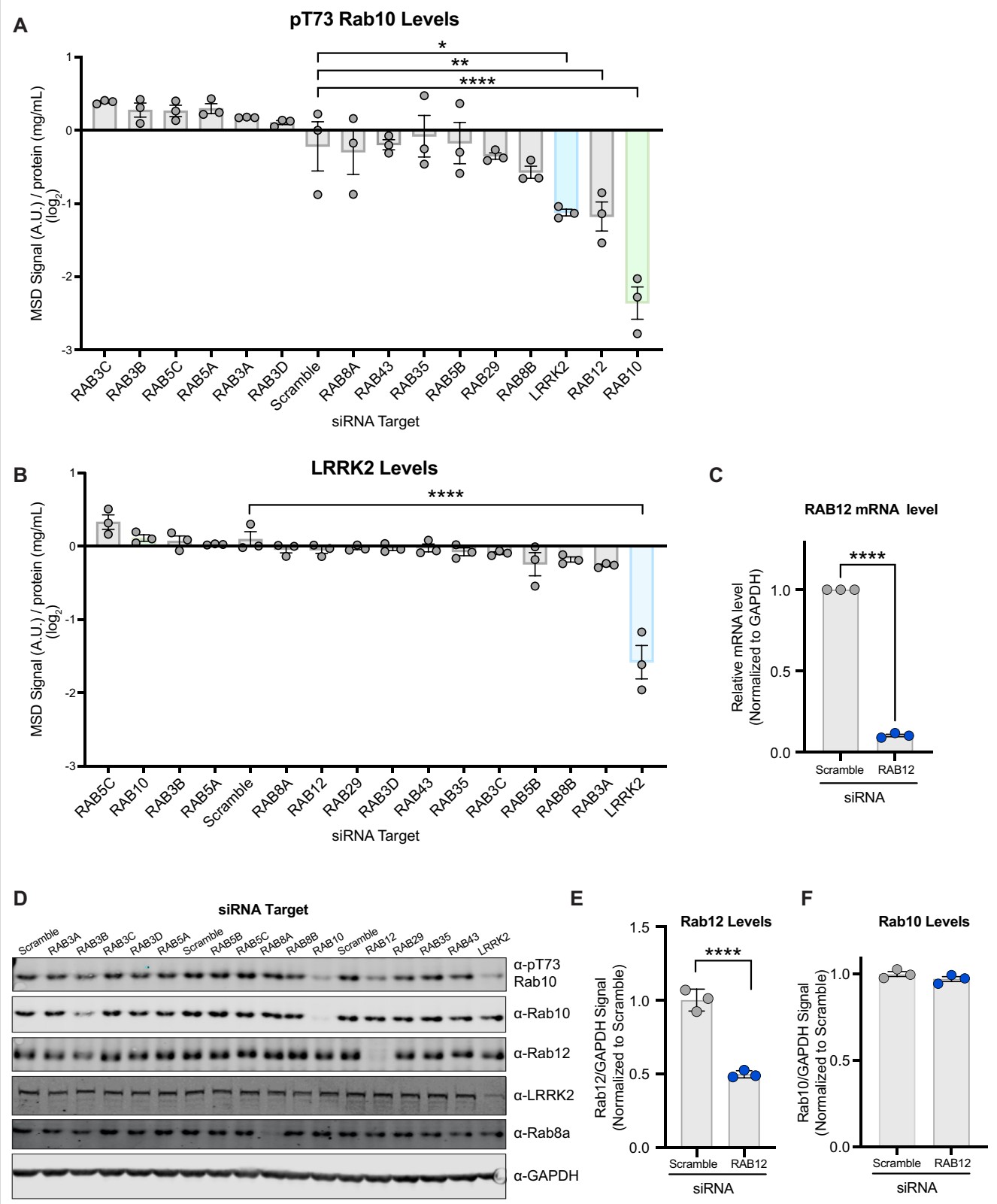

**Figure 1.** A targeted siRNA screen identifies Rab12 as a key regulator of LRRK2 kinase activity. (**A and B**) A549 cells were transfected with siRNA targeting LRRK2 and its Rab substrates, lysed 3 days after transfection, and the levels of pT73 Rab10 and LRRK2 were quantified using Meso Scale Discovery (MSD)-based analysis. The MSD signal was normalized to the protein concentration, and data are shown on a $\log_2$ scale as the mean ± SEM; n=3 independent experiments, and statistical significance was determined using one-way analysis of variance (ANOVA) with Dunnett's multiple

*Figure 1 continued on next page*

*Figure 1 continued*

comparison test. (**C**) RAB12 mRNA levels were quantified using RT-qPCR-based analysis and normalized to GAPDH following transfection with siRNAs targeting a scramble sequence or RAB12. Data are shown as the mean ± SEM; n=3 independent experiments, and statistical significance was determined using paired t-test. (**D**) The levels of pT73 Rab10, Rab10, Rab12, LRRK2, and Rab8a following siRNA-mediated knockdown of LRRK2 and its Rab substrates were assessed in A549 cells by western blot analysis. Shown is a representative immunoblot with GAPDH as a loading control. (**E and F**) The immunoblot signals from multiple experiments were quantified, and the Rab12 and Rab10 signal was normalized to GAPDH, normalized to the median within each batch and expressed as a fold change compared to the scramble control; data are shown as the mean ± SEM; n=3 independent experiments. Statistical significance was determined using unpaired t-test. *p<0.05, **p<0.01, ****p<0.0001.

The online version of this article includes the following source data and figure supplement(s) for figure 1:

**Source data 1.** Raw data files for western blots.

**Source data 2.** Annotated western blots.

**Figure supplement 1.** Confirmation of siRNA-mediated knockdown of LRRK2 Rab substrates.

**Figure supplement 1—source data 1.** Raw data files for western blots.

**Figure supplement 1—source data 2.** Annotated western blots.

## Rab12 promotes LRRK2 activation by PD-linked genetic variants or lysosomal damage

Previous studies have established that pathogenic PD-linked variants and lysosomal membrane disruption can lead to increased LRRK2 kinase activity (*Bonet-Ponce et al., 2020*; *Eguchi et al., 2018*; *Mir et al., 2018*; *Wang et al., 2021*). We next explored whether Rab12 might mediate LRRK2 activity in the context of either a pathogenic variant in *LRRK2* (R1441G) or *VPS35* (D620N). Rab10 phosphorylation was significantly reduced with RAB12 knockdown in LRRK2 R1441G KI and VPS35 D620N KI A549 cells (*Figure 2G*, *Figure 2—figure supplement 1*). Lysosomal membrane damage also increases LRRK2's kinase activity, as treatment with L-leucyl-L-leucine methyl ester (LLOMe), a lysosomotropic agent that condenses into membranolytic polymers and ruptures the lysosomal membrane, has been shown to increase LRRK2-dependent phosphorylation of its Rab substrates (*Bonet-Ponce et al., 2020*; *Eguchi et al., 2018*). We confirmed that LLOMe treatment led to a significant increase in Rab10 phosphorylation in WT cells and demonstrated this effect was abolished in RAB12 KO cells (*Figure 2H*). Together, these data demonstrate that Rab12 is required to mediate LRRK2 activation in response to specific genetic variants associated with PD and lysosomal stress more broadly.

## Rab12 regulates LRRK2-dependent phosphorylation of Rab10 on lysosomes

Lysosomal membrane permeabilization has been shown to increase the levels of LRRK2 and pT73 Rab10 associated with lysosomes using overexpression systems (*Bonet-Ponce et al., 2020*; *Eguchi et al., 2018*). Our data suggested that Rab12 may play a key role in facilitating the recruitment of LRRK2 and ultimate phosphorylation of Rab10 on lysosomes in response to lysosomal damage. To assess this, we employed an established lysosome immunoprecipitation (Lyso-IP) method that enables the rapid isolation of lysosomes (*Abu-Remaileh et al., 2017*). Lysosomes isolated from WT and RAB12 KO A549 cells treated with vehicle or LLOMe (1 mM) for 2 hr displayed increased levels of endogenous galectin-3 (Gal3), validating that LLOMe treatment induced lysosomal membrane rupture and exposed beta-galactosides normally present in the lumen of lysosomes (*Figure 3A*; *Jia et al., 2020*; *Maejima et al., 2013*; *Paz et al., 2010*). While LLOMe treatment reduced the levels of lysosomal-associated membrane protein 1 (LAMP1) in isolated lysosomes, the levels and localization of TMEM192-3x-HA, the lysosomal membrane protein used to isolate lysosomes, were not significantly impacted by LLOMe treatment (*Figure 3A* and *Figure 3—figure supplement 1*). We did not observe a loss of LAMP1 signal by immunofluorescence analysis, suggesting that LAMP1 may dissociate or be degraded from ruptured lysosomal membranes upon immunopurification. These data suggest that while LLOMe treatment results in lysosomal membrane damage, sufficient lysosomal integrity remains to enable purification of this subcellular compartment using TMEM192.

Using this method, we showed that LLOMe treatment increased phosphorylation of Rab10 on isolated lysosomes from WT cells but failed to increase Rab10 phosphorylation on lysosomes from RAB12 KO cells, demonstrating that Rab12 is a critical regulator of Rab10 phosphorylation on lysosomes following lysosomal damage (*Figure 3A* and *Figure 3—figure supplement 1*). While Rab10

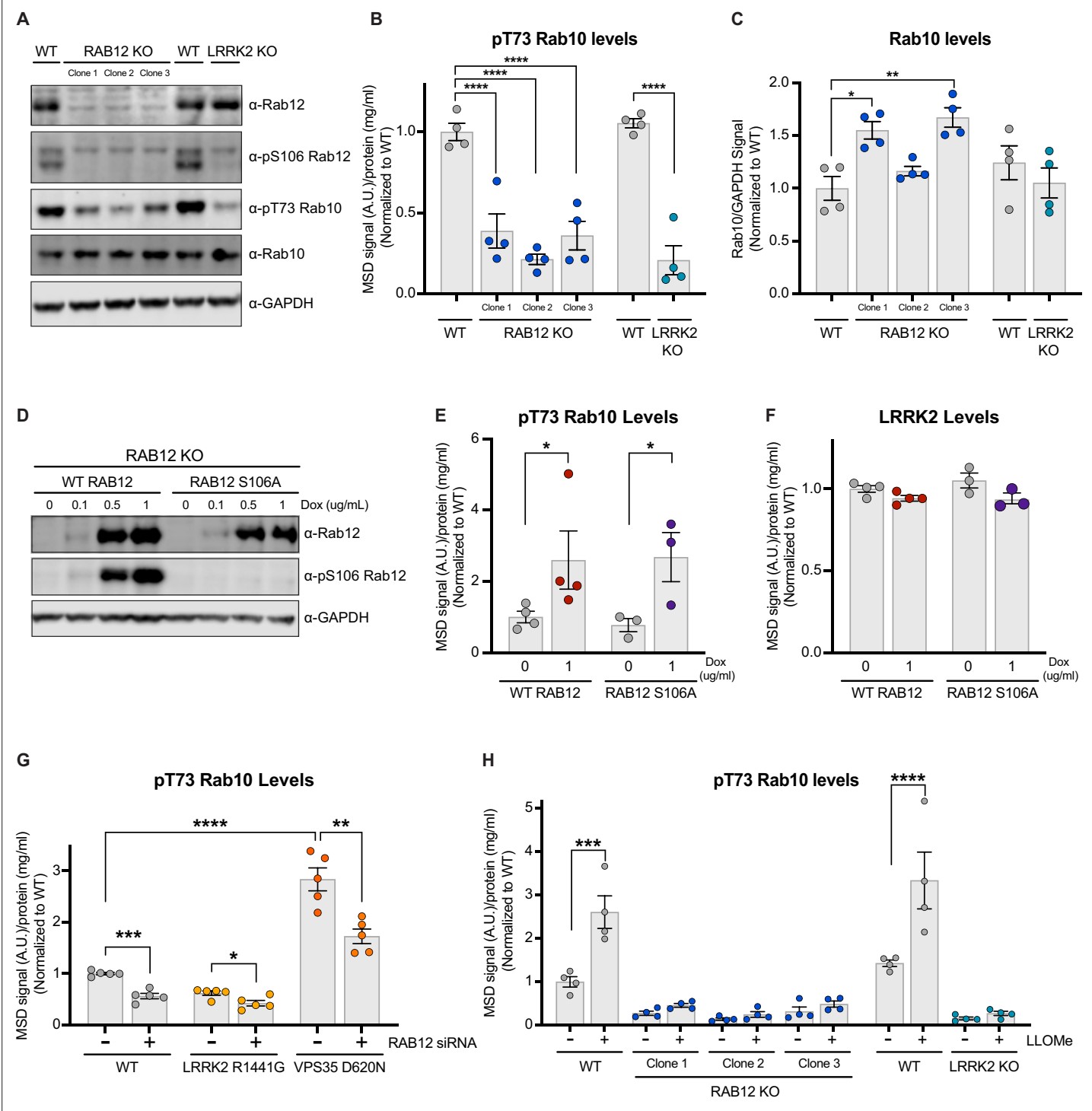

**Figure 2.** Rab12 regulates LRRK2 activation basally and in response to lysosomal damage and genetic variants associated with Parkinson's disease (PD). (**A**) The levels of Rab12, pS106 Rab12, pT73 Rab10, and Rab10 were assessed in wildtype (WT), RAB12 KO, and LRRK2 KO A549 cells by western blot analysis. Shown is a representative immunoblot with GAPDH as a loading control. (**B**) The levels of pT73 Rab10 were measured using a Meso Scale Discovery (MSD)-based assay. The MSD signal was normalized for protein input and expressed as a fold change compared to WT A549 cells; data are shown as the mean ± SEM; n=4 independent experiments, and statistical significance was determined using one-way analysis of variance (ANOVA) with Dunnett's multiple comparison test. (**C**) Immunoblot signals from multiple experiments were quantified, and the Rab10 signal was normalized to GAPDH and expressed as a fold change compared to WT A549 cells. Data are shown as the mean ± SEM; n=4 independent experiments. Statistical significance was determined using one-way ANOVA with Dunnett's multiple comparison test. (**D–F**) RAB12 KO A549 cells with doxycycline-inducible expression of WT RAB12 or a phospho-deficient variant of RAB12 (S106A) were treated with increasing concentrations of doxycycline for 3 days, and the

*Figure 2 continued on next page*

*Figure 2 continued*

levels of Rab12, pS106 Rab12, pT73 Rab10, and LRRK2 were measured. (**D**) A representative immunoblot is shown assessing Rab12 and pS106 Rab12 protein levels following doxycycline-induced expression of WT or RAB12 S106A, and GAPDH was used as a loading control. (**E and F**) The levels of pT73 Rab10 and LRRK2 were measured using MSD-based assays. MSD signals were normalized for protein concentration, and data were then normalized to the median within each batch and to the signals from the control group (RAB12 KO cells with inducible expression of WT Rab12 without doxycycline treatment). Data are shown as mean ± SEM; n=3–4 independent experiments, and statistical significance was determined using unpaired t-test on log transformed data. (**G**) The impact of Rab12 knockdown was measured in WT, LRRK2 R1441G KI, and VPS35 D620N KI A549 cells. Cells were transfected with siRNA targeting RAB12, and pT73 Rab10 levels were measured by MSD-based analysis 3 days after transfection. The MSD signal was normalized for protein input and then normalized to the median within each batch and is expressed as a fold change compared to WT A549 cells transfected with scramble siRNA. Data are shown as the mean ± SEM; n=5 independent experiments. Statistical significance was determined using one-way ANOVA with Tukey's multiple comparison test on log transformed data. (**H**) WT, RAB12 KO, and LRRK2 KO A549 cells were treated with vehicle or L-leucyl-L-leucine methyl ester (LLOMe) (1 mM) for 2 hr, and the impact of LLOMe treatment on pT73 Rab10 levels was measured by MSD-based analysis. The MSD signal was normalized for protein input and is expressed as a fold change compared to WT A549 cells treated with vehicle. Data are shown as the mean ± SEM; n=4 independent experiments. Statistical significance was determined using two-way ANOVA with Sidak's multiple comparison test. *p<0.05, **p<0.01, ***p<0.001, ****p<0.0001.

The online version of this article includes the following source data and figure supplement(s) for figure 2:

**Source data 1.** Raw data files for western blot.

**Source data 2.** Annotated western blots.

**Figure supplement 1.** Confirmation of effects of RAB12 KO and RAB12 S106A KI on total and phospho-Rab12 and Rab10.

**Figure supplement 1—source data 1.** Raw data files for western blot.

**Figure supplement 1—source data 2.** Annotated western blots.

has been reported to primarily localize to the Golgi and endosomes, our data show that a proportion of Rab10 is localized to lysosomes basally and in response to lysosomal damage (*Berndsen et al., 2019*; *Wang et al., 2010*). To further explore the effect of Rab12 on the lysosomal levels of phospho-Rab10, we visualized phosphorylated Rab10 on lysosomes following LLOMe treatment in WT and RAB12 KO cells. LLOMe treatment significantly increased colocalization between pT73 Rab10 and LAMP1 in WT cells but had no effect in RAB12 KO cells, confirming that Rab12 is required for Rab10 phosphorylation on lysosomes in response to membrane rupture (*Figure 3B* and *Figure 3—figure supplement 1*).

## Rab12 increases Rab10 phosphorylation by facilitating LRRK2 recruitment to lysosomes

We hypothesized that lysosomal membrane permeabilization may increase Rab12 recruitment to damaged lysosomes and that increased Rab12 levels on lysosomes may facilitate the lysosomal association of LRRK2 upon damage. Consistent with this idea, treatment with LLOMe significantly increased the levels of Rab12 on lysosomes assessed by western blot analysis from isolated lysosomes and by confocal imaging (*Figure 3C–E* and *Figure 3—figure supplement 1*). Further imaging analysis revealed that LLOMe treatment significantly increased the colocalization of Rab12 with the lysosomal marker LAMP1, but not with the Golgi marker GM130, as quantified by Pearson's correlation coefficient (PCC), and this colocalization was preserved upon nocodazole-induced microtubule depolymerization (*Figure 3D* and *Figure 3—figure supplements 1 and 2*). Rab12 recruitment to damaged lysosomes was not impacted by LRRK2 deletion as Rab12 levels were similarly increased on lysosomes from WT and LRRK2 KO cells following lysosomal membrane permeabilization (*Figure 3E* and *Figure 3—figure supplement 2*). These data demonstrate that lysosomal membrane damage increases Rab12 localization to lysosomes and that this occurs in an LRRK2-independent manner, supporting the idea that Rab12 is an upstream regulator of LRRK2's lysosomal localization and activity. Western blot analysis of isolated lysosomes showed that approximately 1% of total Rab12 was present on lysosomes at baseline and that this increased to approximately 1.5% following LLOMe treatment, while our imaging-based analysis revealed that approximately 12% of overexpressed Rab12 was present on lysosomes at baseline and increased to approximately 14% upon LLOMe treatment (*Figure 3—figure supplement 2*). These results suggest that a small percentage of Rab12 and LRRK2 are present on lysosomes at baseline and that lysosomal damage leads to a significant increase in the localization of both proteins to the lysosome (*Figure 3—figure supplement 2*), but precise quantification of the amount of Rab12 and LRRK2 on lysosomes under these conditions is difficult and warrants further study.

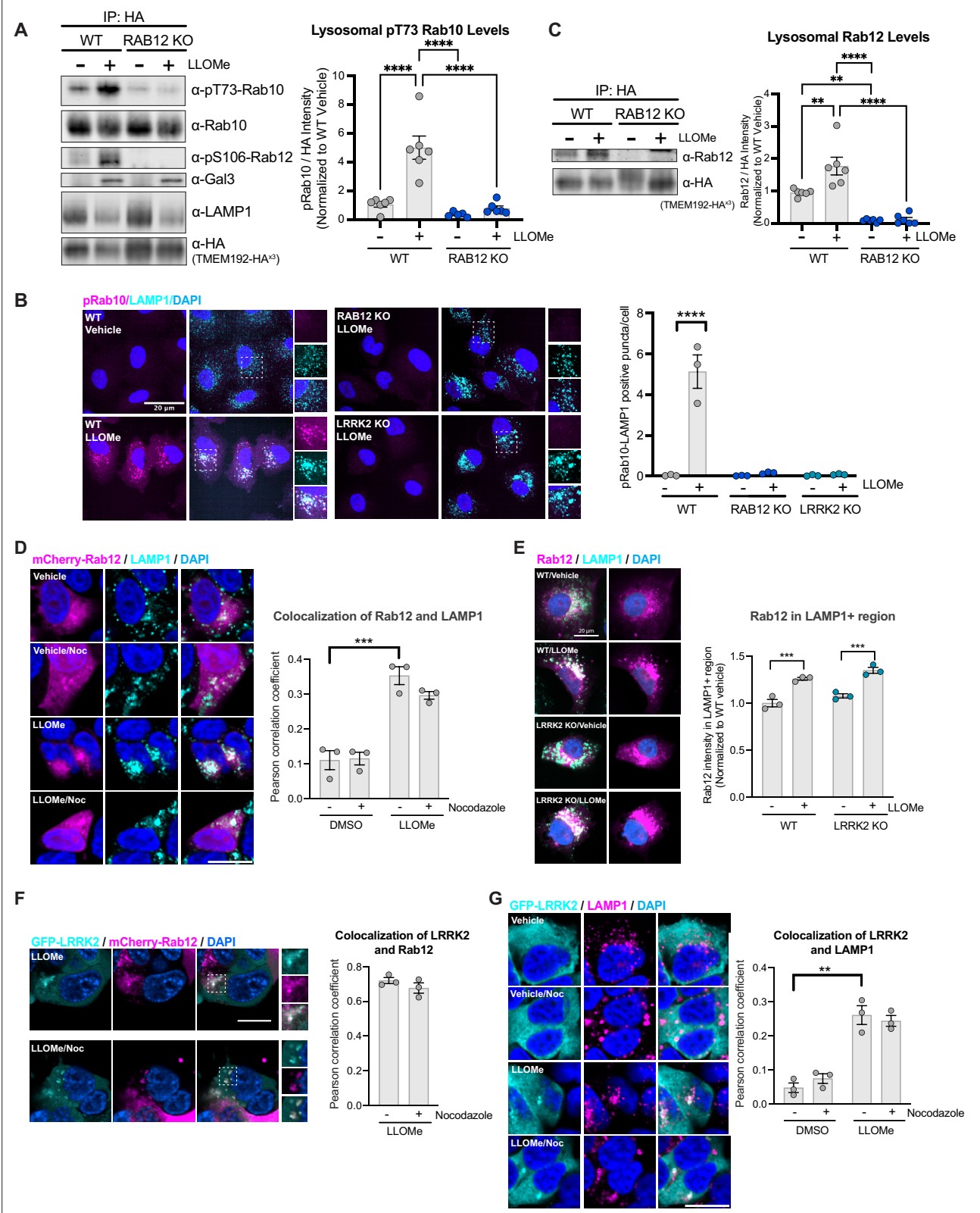

**Figure 3.** Rab12 is recruited to lysosomes following lysosomal damage and promotes Rab10 phosphorylation at the lysosome. (**A**) Lysosomes were isolated from wildtype (WT) and RAB12 KO A549 cells treated with vehicle or L-leucyl-L-leucine methyl ester (LLOMe) (1 mM) for 2 hr. The levels of pT73 Rab10, total Rab10, pS106 Rab12, galectin-3 (Gal3), lysosomal-associated membrane protein 1 (LAMP1), and HA were assessed by western blot analysis, and shown is a representative immunoblot. Fluorescence signals of immunoblots from multiple experiments were quantified. The pT73 Rab10 signal

*Figure 3 continued*

was normalized to the HA signal, then was normalized to the median within each experimental replicate and expressed as a fold change compared to lysosomes isolated from WT A549 cells treated with vehicle. n=6 independent experiments. Data are shown as the mean ± SEM, and statistical significance was determined using one-way analysis of variance (ANOVA) with Tukey's multiple comparison test. (**B**) WT, RAB12 KO, and LRRK2 KO A549 cells were treated with vehicle or LLOMe (1 mM) for 2 hr, and the signals of pT73 Rab10 and LAMP1 were assessed by immunostaining. Scale bar, 20 µm. pT73 Rab10 (shown in magenta) and LAMP1 (shown in cyan) double positive puncta (i.e. overlap of magenta and cyan and shown in white) were quantified per cell from n=3 independent experiments. Data are shown as the mean ± SEM with and statistical significance was determined using two-way ANOVA with Sidak's multiple comparison test. (**C**) Lysosomal Rab12 levels were assessed by western blot analysis from lysosomes isolated from WT and RAB12 KO A549 cells treated with vehicle or LLOMe (1 mM) for 2 hr. The Rab12 signals were normalized to the HA signals, then were normalized to the median within each experimental replicate and expressed as a fold change compared to lysosomes isolated from WT A549 cells treated with vehicle. n=6 independent experiments. Data are shown as the mean ± SEM, and statistical significance was determined using one-way ANOVA with Tukey's multiple comparison test. (**D**) HEK293T cells expressing mCherry-Rab12 were treated with vehicle or LLOMe (1 mM) for 2 hr, fixed, and stained using an antibody against LAMP1. Colocalization of Rab12 and LAMP1 was assessed by measuring the Pearson's correlation coefficient between mCherry-Rab12 (shown in magenta) and LAMP1 (shown in cyan); nocodazole (25 µM for 2 hr) treatment was included as a control to confirm colocalization. Scale bar, 10 µm. n=3 independent experiments. Data are shown as the mean ± SEM, and statistical significance was determined using repeated measures one-way ANOVA with Sidak's multiple comparison test. (**E**) WT and LRRK2 KO A549 cells transiently expressing mCherry-Rab12 were treated with vehicle or LLOMe (1 mM) for 2 hr, and the LAMP1 levels were assessed by immunostaining. Scale bar, 20 µm. The intensity of mCherry-Rab12 signals (shown in magenta) in LAMP1 (shown in cyan)-positive region were quantified per cell from mCherry-Rab12 expressing cells (n=20 cells per condition, with cellular intensity between 2000 and 5000 fl. units) and averaged across wells (~4–6 wells per condition). n=3 independent experiments. The Rab12 signal was normalized to the median within each experimental replicate, and then expressed as a fold change compared to WT cells treated with vehicle. Data are shown as the mean ± SEM, and statistical significance was determined using one-way ANOVA with Sidak's multiple comparison test. (**F**) HEK293T cells stably expressing eGFP-LRRK2 were transfected with mCherry-Rab12 and treated with LLOMe (1 mM) for 2 hr. Colocalization of mCherry-Rab12 (shown in magenta) and eGFP-LRRK2 (shown in cyan) was assessed by measuring the Pearson's correlation coefficient in LLOMe-responding cells (n=10 cells per condition); nocodazole (25 µM) treatment was included to confirm colocalization. Scale bar, 10 µm. n=3 independent experiments. (**G**) HEK293T cells stably expressing eGFP-LRRK2 were treated with vehicle or LLOMe (1 mM) for 2 hr, fixed, and stained using an antibody against LAMP1. Colocalization of LRRK2 and LAMP1 was assessed by measuring the Pearson's correlation coefficient between eGFP-LRRK2 (shown in cyan) and LAMP1 (shown in magenta); nocodazole (25 µM) treatment was included to confirm colocalization. Scale bar, 10 µm. n=3 independent experiments. Data are shown as the mean ± SEM, and statistical significance was determined using repeated measures one-way ANOVA with Sidak's multiple comparison test. **p<0.01, ***p<0.001, and ****p<0.0001.

The online version of this article includes the following source data and figure supplement(s) for figure 3:

**Source data 1.** Raw data files for western blot.

**Source data 2.** Annotated western blots.

**Figure supplement 1.** Validation of lysosomal immunopurification method and analysis of pT73 Rab10 and Rab12 localization in response to lysosomal damage.

**Figure supplement 1—source data 1.** Raw data files for western blot.

**Figure supplement 1—source data 2.** Annotated western blots.

**Figure supplement 2.** Analysis of Rab12 and LRRK2 localization at baseline and in response to lysosomal damage.

To gain additional insight around the dynamics of Rab12 and LRRK2 recruitment following lysosomal membrane permeabilization, we performed live-cell imaging of HEK293T cells overexpressing mCherry-tagged Rab12 and eGFP-tagged LRRK2 and assessed Rab12 and LRRK2 localization over time. Rab12 and LRRK2 showed a diffuse localization under baseline conditions, while LLOMe treatment increased the recruitment of Rab12 and LRRK2 to vesicular structures (*Figure 3—figure supplement 2*). Rab12 colocalization with LRRK2 increased over time following LLOMe treatment, supporting potential coordinated recruitment of these proteins to lysosomes upon damage (*Figure 3F* and *Figure 3—figure supplement 2*). LLOMe treatment also significantly increased the colocalization of LRRK2 with the lysosomal marker LAMP1 but not with the Golgi marker GM130 (*Figure 3G* and *Figure 3—figure supplement 2*). Together, these data demonstrate that Rab12 and LRRK2 both associate with lysosomes following membrane rupture.

To more directly assess whether Rab12 regulates the recruitment of LRRK2 to lysosomes upon lysosomal damage, we next examined the impact of RAB12 deletion on the lysosomal recruitment of LRRK2 upon lysosomal membrane permeabilization. Lysosomes were isolated from WT and RAB12 KO cells, and the endogenous levels of LRRK2 on lysosomes were quantified by western blot analysis. Lysosomal levels of LRRK2 were increased by approximately 2.5-fold following treatment with LLOMe in WT cells, and loss of Rab12 abrogated this LLOMe-induced increase (*Figure 4A*). These data show that Rab12 is required to facilitate LRRK2 localization to lysosomes following membrane damage.

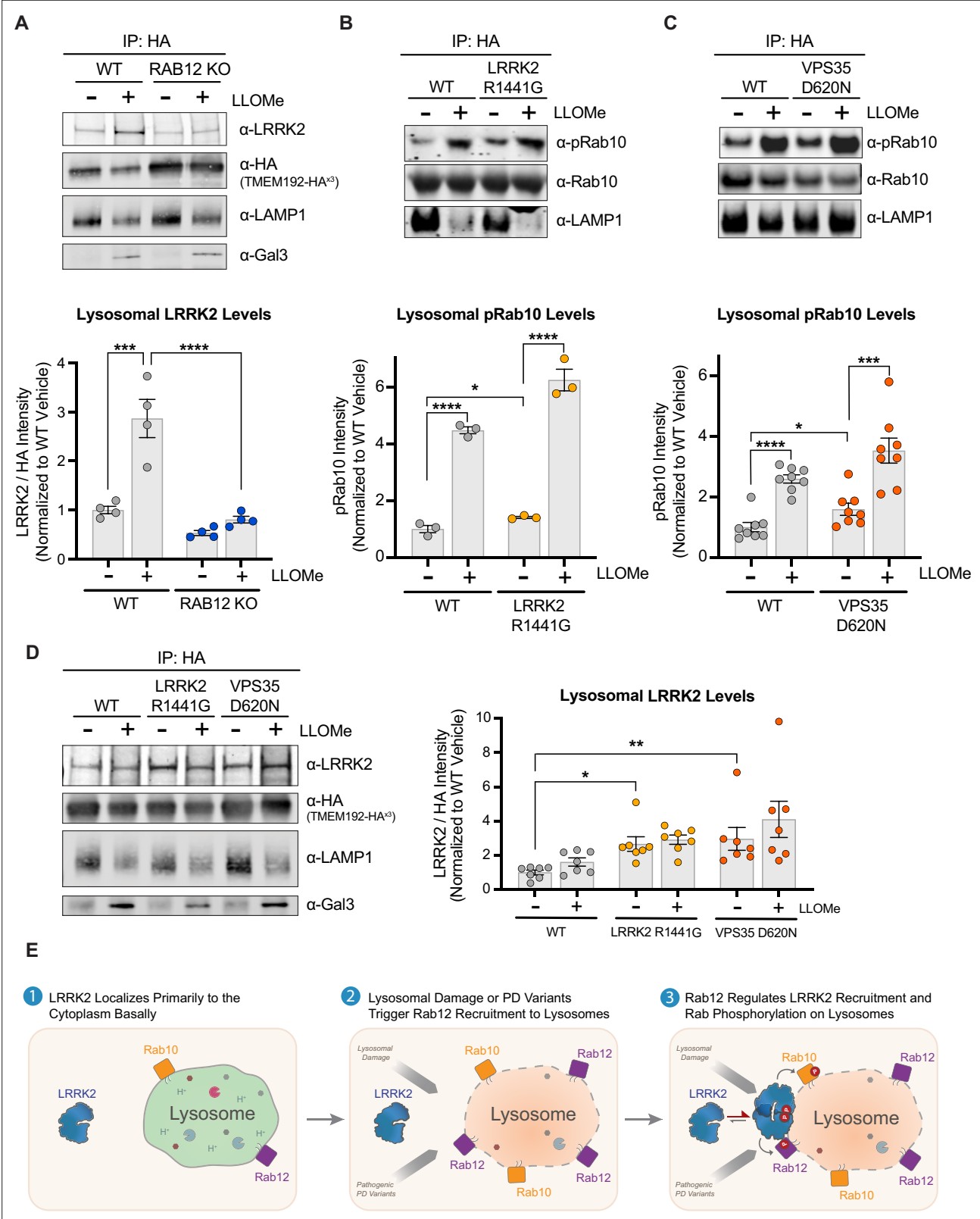

**Figure 4.** LRRK2 levels are increased on lysosomes following lysosomal damage in a Rab12-dependent manner and are also increased by Parkinson's disease (PD)-linked variants. (**A**) To analyze lysosomal LRRK2 levels, lysosomes were isolated from wildtype (WT) and RAB12 KO A549 cells treated with vehicle or L-leucyl-L-leucine methyl ester (LLOMe) (1 mM) for 4 hr. The levels of LRRK2, HA, lysosomal-associated membrane protein 1 (LAMP1), and galectin-3 (Gal3) were assessed by western blot analysis, and shown is a representative immunoblot. Fluorescence signals of immunoblots from multiple

*Figure 4 continued on next page*

*Figure 4 continued*

experiments were quantified, LRRK2 signal was normalized to the HA signal, then normalized to the median within each experiment, and expressed as a fold change compared to lysosomes isolated from WT A549 cells treated with vehicle. Data are shown as the mean ± SEM; n=4 independent experiments. Statistical significance was determined using one-way analysis of variance (ANOVA) with Tukey's multiple comparison test with a single pooled variance. (**B–C**) To analyze lysosomal pRab10 levels, lysosomes were isolated from WT and LRRK2 R1441G KI (**B**) or VPS35 D620N KI (**C**) A549 cells treated with vehicle or LLOMe (1 mM) for 2 hr, and the levels of pT73 Rab10, Rab10, and LAMP1 were assessed by western blot analysis. Immunoblot signals from multiple experiments were quantified, and the pT73 Rab10 signal was expressed as a fold change compared to lysosomes isolated from WT A549 cells treated with vehicle. Data are shown as the mean ± SEM; n=3 independent experiments (**B**) and n=8 independent experiments (**C**). Statistical significance was determined using unpaired t-test. (**D**) Lysosomes were isolated from WT, LRRK2 R1441G KI, and VPS35 D620N KI A549 cells treated with vehicle or LLOMe (1 mM) for 4 hr. The levels of LRRK2, HA, LAMP1, and Gal3 were assessed by western blot analysis and shown is a representative immunoblot. Fluorescence signals of immunoblots from multiple experiments were quantified, the LRRK2 signal was normalized to the HA signal, then normalized to the median within each experiment, and expressed as a fold change compared to lysosomes isolated from WT A549 cells treated with vehicle. Data are shown as the mean ± SEM; n=7 independent experiments. Statistical significance was determined using one-way ANOVA with Dunnett's multiple comparison test. *p<0.05, ***p<0.001, ****p<0.0001. (**E**) Model for proposed mechanism by which Rab12 promotes LRRK2 activation. Under steady-state conditions, LRRK2 localizes primarily to the cytoplasm. Lysosomal damage prompts the recruitment of Rab12, and Rab12 regulates the recruitment of LRRK2 to damaged lysosomes. An elevated local concentration of LRRK2 on lysosomes increases the likelihood for interactions with Rab GTPases localized on the lysosomal membrane, promoting LRRK2-dependent phosphorylation of its Rab substrates.

The online version of this article includes the following source data and figure supplement(s) for figure 4:

**Source data 1.** Raw data files for western blot.

**Source data 2.** Annotated western blots.

**Figure supplement 1.** Western blot analysis of pRab10 and LRRK2 levels on isolated lysosomes and the corresponding post-nuclear supernatant (PNS).

Enhanced recruitment of LRRK2 to lysosomes may promote Rab phosphorylation by effectively increasing the local concentration of LRRK2 in proximity to its Rab substrates, and we hypothesized that such a mechanism might explain LRRK2 activation observed in additional contexts beyond lysosomal damage. We next examined whether two PD-linked genetic variants associated with increased LRRK2 kinase activity, LRRK2 R1441G and VPS35 D620N, also had increased levels of LRRK2 on lysosomes. Lysosomes were isolated from WT, LRRK2 R1441G KI, and VPS35 D620N KI A549 cells at baseline and following LLOMe treatment, and the levels of total and phospho-Rab10 and LRRK2 were subsequently assessed by western blot analysis. Expression of LRRK2 R1441G or VPS35 D620N led to an increase in Rab10 phosphorylation on isolated lysosomes at baseline, and the phosphorylation of Rab10 on lysosomes was further increased following LLOMe treatment (*Figure 4B and C*). The levels of LRRK2 on lysosomes were significantly increased in untreated LRRK2 R1441G KI cells and VPS35 D620N KI cells, suggesting that enhanced localization of LRRK2 to lysosomes and proximity to its Rab substrates may contribute to the elevated Rab10 phosphorylation observed on lysosomes at baseline in these cells (*Figure 4D*). Rab10 phosphorylation was increased on lysosomes in response to LLOMe treatment while the levels of LRRK2 were not significantly impacted on lysosomes isolated from LRRK2 R1441G and VPS35 D620N KI cells, suggesting additional mechanisms beyond LRRK2 localization may also contribute to LRRK2 activation in response to lysosomal damage in these cells. Together, these results suggest that Rab12 regulates LRRK2 localization to lysosomes upon damage and that this may be a conserved mechanism also employed to contribute to LRRK2 activation in response to PD-linked variants.

## Discussion

Increased LRRK2 kinase activity is observed with pathogenic PD-linked variants and in sporadic PD patients, but many questions remain regarding the mechanisms by which LRRK2 activation is regulated basally and in response to endolysosomal stress associated with disease. We identify Rab12 as a novel regulator of LRRK2 activation and demonstrate that Rab12 plays a critical role in mediating LRRK2-dependent Rab phosphorylation in response to lysosomal damage. A recent CRISPR-based genome-wide screen for modifiers of Rab10 phosphorylation also identified RAB12 as a top hit, providing additional validation that Rab12 is a key regulator of LRRK2 activation (*Dhekne et al., 2023*). Our data show that lysosomal membrane rupture promotes Rab12 localization to lysosomes. We propose a model in which Rab12 recruits LRRK2 to the lysosome and enhances its activity on lysosomal membranes by increasing LRRK2's local concentration near Rab10 and potentially other Rab

substates (*Figure 4E*). Previous studies suggested that another LRRK2 substrate, Rab29, can regulate LRRK2's kinase activity and showed that exogenously expressed Rab29 is capable of activating LRRK2 at the trans-Golgi or at additional membranes when artificially anchored there (*Purlyte et al., 2018*; *Gomez et al., 2019*). However, these studies relied on overexpression of Rab29 to increase LRRK2 activity, and analyses of RAB29 KO cellular models or mice showed that RAB29 deletion minimally impacted LRRK2-dependent Rab10 phosphorylation (*Kalogeropulou et al., 2020*). Here, we used endogenous expression conditions to demonstrate that Rab12, but not Rab29, regulates LRRK2-mediated phosphorylation of Rab10 and that Rab12 regulates the localization and activation of LRRK2 on lysosomes upon lysosomal stress. The PD-linked pathogenic LRRK2 variant R1441G, which lies outside of LRRK2's kinase domain, and the VPS35 D620N variant have been shown in previous studies and confirmed here to increase LRRK2 activity (*Liu et al., 2018*; *Mir et al., 2018*; *Purlyte et al., 2018*). Our results suggest that these variants may also promote LRRK2 activation by increasing Rab12-mediated LRRK2 localization to lysosomes, implying this may represent a general mechanism by which LRRK2 activity is increased by various genetic variants and environmental stressors associated with PD. We cannot rule out the possibility that additional mechanisms beyond increased proximity between LRRK2 and its Rab substrates may contribute to Rab10 phosphorylation following lysosomal damage, as the magnitude of change in Rab10 phosphorylation induced by LLOMe treatment was greater than the extent of LRRK2 recruitment to damaged lysosomes. Additional studies are warranted to determine how lysosomal membrane rupture triggers Rab12 recruitment, to identify other regulatory processes that may contribute to Rab phosphorylation upon lysosomal damage, and to better define how broadly such mechanisms are employed to drive LRRK2 activation in PD. As our work focused on the role of Rab12 in A549 cells, it will also be important to understand whether Rab12 similarly regulates LRRK2 activation and localization to damaged lysosomes in other cell types.

Our findings provide key insight into the mechanism by which LRRK2 activity is increased in response to lysosomal damage by demonstrating that Rab12 regulates LRRK2 localization to ruptured lysosomes. While the purpose of LRRK2 recruitment and activation on lysosomes is poorly defined, several hypotheses have been proposed suggesting that LRRK2 activity may be upregulated as a compensatory response to restore lysosomal homeostasis following damage. Previous work supports a model in which LRRK2 activity controls the decision to repair minor damage to lysosomal membranes through an ESCRT-mediated process or to target damaged lysosomes for degradation via lysophagy (*Herbst et al., 2020*). Additional studies suggested that LRRK2 hyperactivation triggers mechanisms aimed at clearing cargo from damaged lysosomes that have lost their proteolytic capacity, either through direct exocytosis of lysosomes or through a novel sorting process termed LYTL (lysosomal tubulation/sorting driven by LRRK2) in which tubules bud off from lysosomes following membrane rupture (*Bonet-Ponce et al., 2020*; *Eguchi et al., 2018*; *Kluss et al., 2022*). LRRK2 activation in the face of minor or acute lysosomal insults likely plays a beneficial role in maintaining lysosomal function, while chronic LRRK2 activation triggered by genetic variants with increased kinase activity or low-level lysosomal damage over time may ultimately impair the ability to effectively respond to membrane stress and maintain the balance between lysosomal repair and destruction. Macrophages from PD patients carrying pathogenic LRRK2 variants were shown to accumulate more damaged lysosomes compared to samples from healthy controls, suggesting that mechanisms that respond to and repair lysosomal membrane rupture may be perturbed in PD (*Herbst et al., 2020*). A deeper understanding of the mechanisms by which LRRK2 responds to lysosomal damage and how these contribute to PD pathogenesis is critical in guiding new potential therapeutic strategies targeting LRRK2 for the treatment of PD.

## Materials and methods

**Key resources table**

| Reagent type (species) or resource | Designation | Source or reference | Identifiers | Additional information |
|---|---|---|---|---|
| Antibody | pT73 Rab10 (rabbit monoclonal) | Denali | 19-4 | WB: 1:500; MSD 1 ug/mL |
| Antibody | pT73 Rab10 (rabbit monoclonal) | Abcam | ab241060 RRID:AB_2884876 | ICC: 1:100 |

*Continued on next page*

*Continued*

| Reagent type (species) or resource | Designation | Source or reference | Identifiers | Additional information |
|---|---|---|---|---|
| Antibody | Rab10 (mouse monoclonal) | Abcam | ab104859 RRID:AB_10711207 | WB: 1:500 |
| Antibody | Rab10 (rabbit monoclonal) | CST | 8127 RRID:AB_10828219 | ICC: 1:100 |
| Antibody | Rab10 (rabbit monoclonal) | Abcam | ab181367 | MSD: 2 ug/mL |
| Antibody | pS106 Rab12 (rabbit monoclonal) | Abcam | ab256487 RRID:AB_2884880 | WB: 1:500 |
| Antibody | Rab12 (rabbit polyclonal) | Proteintech | 18843–1-AP RRID:AB_10603469 | WB: 1:500 |
| Antibody | Rab8a (rabbit monoclonal) | Abcam | ab188574 RRID:AB_2814989 | WB: 1:500 |
| Antibody | HA (rabbit monoclonal) | CST | 3724 RRID:AB_1549585 | WB: 1:2000 ICC: 1:100 |
| Antibody | Gal3 (mouse monoclonal) | BD | 556904 RRID:AB_396531 | WB: 1:500 |
| Antibody | LAMP1 (rabbit monoclonal) | CST | 9091 RRID:AB_2687579 | WB: 1:500 |
| Antibody | LAMP1 (mouse monoclonal) | Abcam | ab25630 RRID:AB_470708 | WB: 1:1000 ICC: 1:100 |
| Antibody | LRRK2 (mouse monoclonal) | UC Davis/NIH NeuroMab Facility | N241A/34 RRID:AB_2877351 | WB: 1:500 |
| Antibody | LRRK2 (mouse monoclonal) | Biolegend | 808201 RRID:AB_2564739 | MSD: 1 µg/mL |
| Antibody | LRRK2 | Biolegend | 844401 RRID:AB_2565614 | MSD: 0.5 µg/mL |
| Antibody | pS935 LRRK2 (rabbit monoclonal) | Abcam | ab133450 RRID:AB_2732035 | WB: 1:500 MSD: 0.5 µg/mL |
| Antibody | GAPDH (mouse monoclonal) | Abcam | ab8245 RRID:AB_2107448 | WB: 1:500 |
| Antibody | Actin (mouse monoclonal) | Sigma-Aldrich | A2228 RRID:AB_476697 | WB: 1:5000 |
| Antibody | GM130 (rabbit monoclonal) | Abcam | ab52649 RRID: AB_880266 | ICC: 1:200 |
| Cell line (*Homo sapiens*) | HEK293T cells | ATCC | CRL-3216 | |
| Cell line (*H. sapiens*) | A549 cells | ATCC | CCL-185 | |
| Cell line (*H. sapiens*) | Rab29 KO A549 cells | Dr. Dario Alessi | | |
| Cell line (*H. sapiens*) | Rab12 KO A549 cells | Denali | | |
| Cell line (*H. sapiens*) | LRRK2 R1441G KI A549 cells | Denali | | |
| Cell line (*H. sapiens*) | VPS35 D620N KI A549 cells | Denali | | |
| Cell line (*H. sapiens*) | eGFP-LRRK2 HEK293T cells | Denali | | |
| Recombinant DNA reagent | pLVX-TMEM192-3x-HA | Denali | | |
| Recombinant DNA reagent | pcDNA3.1-eGFP-LRRK2 | Denali | | |
| Recombinant DNA reagent | pcDNA3.1-mCherry-Rab12 | Denali | | |
| Chemical compound | LLOMe | Sigma-Aldrich | L7393 | 1 mM |
| Chemical compound | Nocodazole | Sigma-Aldrich | M1404 | 25 µM |

*Continued on next page*

*Continued*

| Reagent type (species) or resource | Designation | Source or reference | Identifiers | Additional information |
|---|---|---|---|---|
| Software, algorithm | Fiji | PMID:29187165 | RRID:SCR_002285 | |
| Software, algorithm | GraphPad Prism | Version 9.5.1 | RRID:SCR_002798 | |
| Software, algorithm | Image Studio Lite | Version 5.2.5 | RRID:SCR_013715 | |
| Software, algorithm | Harmony | Versions 4.9 and 5.1 | RRID:SCR_018809 | |
| Software, algorithm | Python Programming Language | v3.10.8 | RRID:SCR_008394 | |
| Software, algorithm | SciPy | v1.9.3 | RRID:SCR_008058 | |
| Software, algorithm | Napari | v0.4.17 | RRID:SCR_022765 | |
| Software, algorithm | scikit-image | v0.19.3 | RRID:SCR_021142 | |
| Software, algorithm | NumPy | v1.24.2 | RRID:SCR_008633 | |

## Generation of CRISPR KO and KI cell lines

Cell line engineering of A549 cells to generate homozygous LRRK2 R1441G (CGC/GGC) knock-in, homozygous LRRK2 KO, homozygous RAB12 KO, and homozygous VPS35 D620N (GAT/ATT) knock-in was performed using CRISPR/Cas9. Sequence information for generating targeting gRNA, ssODN donor, and PCR primers are as follows:

| LRRK2 R1441G knock-in (A549) | Sequences |
|---|---|
| sgRNA | AAGAAGAAGCGCGAGCCUGG |
| Donor sequence | AAATGTGTGCCAACGAGAATCACA GGGGAAGAAGAAGCGCCAGCCTGG AGGGAAAGACACAAAACCCTCTTGTGTTTGCTTTCAAA |
| Forward PCR primer (5'–3') | AGGCATGAAGATGGGAAAGGA |
| Reverse PCR primer (5'–3') | GGAACCCTCGCTTATTCAGGA |
| **LRRK2 knock-out (A549)** | **Sequences** |
| sgRNA 1 | GGGGACTGTCGACGGTGATCGGT |
| sgRNA 2 | GGTCCTAAACCTGGTCGCAAAGA |
| Donor sequence | n/a |
| Forward PCR primer (5'–3') | AGTCCGCTGAGTCAGTTTCTTC |
| Reverse PCR primer (5'–3') | GGGCTCTTCATCCCGTTTACA |
| **VPS35 D620N knock-in (A549)** | **Sequences** |
| sgRNA | GAUGGCAGCUAGCUGUGCUU |
| Donor sequence | TGTTCACTAGGCATTTTCTCTGTA TGAAGATGAAATCAGCAATTCAAA AGCACAGCTAGCTGCCATCACCTTGATCATTGGCACTTTT GA |
| Forward PCR primer (5'–3') | GGCCATGACAACTGATCCCT |
| Reverse PCR primer (5'–3') | GAGAGGGTGCAGCATGTTCT |
| **Rab12 knock-out (A549)** | **Sequences** |
| sgRNA | AUCAAACUGUAGAGCUAAG |
| Donor sequence | n/a |
| Forward PCR primer (5'–3') | GGGAGGTTATAGACACTGGTGC |
| Reverse PCR primer (5'–3') | AACTGCTCCCCATGTGCAAG |

CRISPR/Cas9-mediated knockout of LRRK2 or RAB12 and CRISPR/Cas9-mediated knock-in of LRRK2 R1441G or VPS35 D620N in A549 cells was performed by Synthego Corporation (Redwood City, CA, USA). To generate these cells, ribonucleoproteins containing the Cas9 protein and synthetic chemically modified sgRNA produced at Synthego were electroporated into the cells using Synthego's optimized protocol. Editing efficiency was assessed upon recovery, 48 hr post electroporation. Specifically, genomic DNA was extracted from a portion of the cells, PCR amplified, and sequenced using Sanger sequencing. The resulting chromatograms are processed using Synthego Inference of CRISPR edits software (ice.synthego.com). To isolate monoclonal cell populations, edited cell pools were seeded at 1 cell/well using a single cell printer into 96- or 384-well plates. All wells were imaged every 3 days to ensure expansion from a single-cell clone. Clonal populations were screened and identified using the PCR-Sanger-ICE genotyping strategy described above.

### Lyso-IP cell line generation

To enable the rapid isolation of lysosomes using immunopurification, WT human A549 cells and different CRISPR A549 cells (including LRRK2 KO, LRRK2 R1441G, RAB12 KO, and VPS35 D620N) were transduced with lentivirus carrying the transgene cassette for expression of TMEM192-3x-HA. A synthetic cDNA encoding TMEM192-3x-HA was cloned into pLVX-IRES-hygromycin lentiviral vector containing the CMV promoter. Stably expressing cells were selected using resistance to Hygromycin B (Thermo Fisher Scientific, Waltham, MA, USA, #10687010) supplied in growth medium at 200 μg/mL for 21 days. Following selection, cells were screened for the stable expression of TMEM192-3x-HA in lysosomes by quantifying the percentage of cells with colocalization of anti-HA and anti-LAMP1 by immunofluorescence, and by monitoring cell lysates for expression TMEM192-3x-HA (~30 kDa) by western blot.

### Antibodies

For ICC, the following secondary antibodies (Thermo Fisher) were used at 1:1000 dilution: goat anti-mouse Alexa-Fluor 488 (A32723), goat anti-rabbit Alexa-Fluor 568 (A11036).

For western blot analysis, the following secondary antibodies (LI-COR Biosciences, Lincoln, NE, USA) were used at 1:20,000 dilution: IRDyes 800CW donkey anti-rabbit IgG (#926-32213), 680RD donkey anti-mouse IgG (#926-68072).

### siRNA-mediated KD of LRRK2 and Rab GTPases

A549 cells were transfected with Dharmacon SMARTpool siRNA targeting 14 Rab GTPases, LRRK2 and non-targeting scramble control (Horizon Discovery, Cambridge, UK), using DharmaFECT 1 (Horizon, T-2001-01). Cells were collected 3 days after transfection for protein or mRNA analysis.

| Targets | Catalog number | Sequence |
|---|---|---|
| ON-TARGETplus Non-targeting Control | D-001810-10 | UGGUUUACAUGUCGACUAA |
| RAB3A | L-009668-00 | GAAGAUGUCCGAGUCGUUG |
| RAB3B | L-008825-00 | GGACACAGACCCGUCGAUG |
| RAB3C | L-008520-00 | UGAGCGAGGUCAACAUUUA |
| RAB3D | L-010822-00 | GUUCAAACUGCUACUGAUA |
| RAB5A | L-004009-00 | GCAAGCAAGUCCUAACAUU |
| RAB5B | L-004010-00 | GGAGCGAUAUCACAGCUUA |
| RAB5C | L-004011-00 | UCAUUGCACUCGCGGGUAA |
| RAB8A | L-003905-00 | CAGGAACGGUUUCGGACGA |
| RAB8B | L-008744-00 | GCAAUUGACUAUGGGAUUA |
| RAB10 | L-010823-00 | GCAAGGGAGCAUGGUAUUA |
| RAB12 | L-023375-02 | CAUUUGAUGAUUUGCCGAA |

*Continued on next page*

*Continued*

| Targets | Catalog number | Sequence |
|---------|----------------|----------|
| RAB29 | L-010556-00 | GAGAACGGUUUCACAGGUU |
| RAB35 | L-009781-00 | GAUGAUGUGUGCCGAAUAU |
| RAB43 | L-028161-01 | GGAUGAGAGGGCACCGCAA |
| LRRK2 | L-006323-00 | GAAAUUAUCAUCCGACUAU |

## RT-qPCR-based analysis of Rab expression

The total RNA was extracted from cells using RNeasy Plus Micro Kit (QIAGEN, Hilden, Germany, #74034). cDNA was synthesized from 1 to 2 µg of RNA using Superscript IV VILO master mix (Thermo Fisher #11756050). The cDNA was diluted threefold and 1 µL of diluted cDNA was used as template. To measure the relative expression levels of mRNAs by RT-qPCR, Taqman Fast Advanced Master Mix (Thermo Fisher #4444557) was used, together with gene-specific primers using TaqMan Assays (Thermo Fisher). GAPDH was used as the housekeeping gene. The PCR was run using QuantStudio 6 Flex Real-Time PCR System, 384-well (Thermo Fisher). Gene expression was analyzed using 2^(delta-delta Ct) method with GAPDH as internal controls.

| Taqman assay ID | Gene name | Dye |
|-----------------|-----------|-----|
| Hs00923221_m1 | RAB3A | FAM-MGB |
| Hs01001137_m1 | RAB3B | FAM-MGB |
| Hs00384846_m1 | RAB3C | FAM-MGB |
| Hs00758197_m1 | RAB3D | FAM-MGB |
| Hs00702360_s1 | RAB5A | FAM-MGB |
| Hs05027271_g1 | RAB5B | FAM-MGB |
| Hs00904926_g1 | RAB5C | FAM-MGB |
| Hs00180479_m1 | RAB8A | FAM-MGB |
| Hs00213006_m1 | RAB8B | FAM-MGB |
| Hs00794658_m1 | RAB10 | FAM-MGB |
| Hs01391604_m1 | RAB12 | FAM-MGB |
| Hs01026316_m1 | RAB29 | FAM-MGB |
| Hs00199284_m1 | RAB35 | FAM-MGB |
| Hs03006628_gH | RAB43 | FAM-MGB |
| Hs01115057_m1 | LRRK2 | FAM-MGB |
| Hs99999905_m1 | GAPDH | VIC |

## MSD-based analysis of pT73 Rab10, total and pSer935 LRRK2

LRRK2, pS935 LRRK2, and pT73-Rab10 MSD assays were previously established (*Wang et al., 2021*). Briefly, capture antibodies were biotinylated using EZ-Link NHS-LC-LC-Biotin (Thermo Fisher, #21343), and detection antibodies were conjugated using Sulfo-TAG NHS-Ester (Meso Scale Discovery [MSD], Rockville, MD, USA, R31AA-1). 96-well MSD GOLD Small Spot Streptavidin plates (MSD, L45SSA-1) were coated with 25 µL of capture antibody diluted in Diluent 100 (MSD, R50AA-2) for 1 hr at room temperature with 700 rpm shaking. After three washes with TBST, 25 µL samples were added each well and incubated at 4°C overnight with agitation at 700 rpm. After three additional washes with TBST, 25 µL of detection antibodies were added to each well diluted in TBST containing 25% MSD blocker A (MSD, R93AA-1) together with rabbit (Rockland Immunochemicals, Pottstown, PA, USA, D610-1000) and mouse gamma globin fraction (Rockland, D609-0100). After

a 1 hr incubation at room temperature at 700 rpm and three washes with TBST, 150 μL MSD read buffer (MSD R92TC, 1:1 diluted with water) was added, and plates were read on the MSD Sector S 600.

| Assay | Antibody type | Targets | Vendor | Catalog number | Concentration (μg/mL) |
|---|---|---|---|---|---|
| | Capture | pS935 LRRK2 | Abcam | ab133450 | 0.5 |
| pS935 LRRK2 | Detection | Total LRRK2 | BioLegend | 808201 | 1 |
| | Capture | Total LRRK2 | BioLegend | 844401 | 0.5 |
| Total LRRK2 | Detection | Total LRRK2 | BioLegend | 808201 | 1 |
| | Capture | pT73 Rab10 | Denali | 19-4 | 1 |
| pT73 Rab10 | Detection | Total Rab10 | Abcam | ab181367 | 2 |

## Cell lysis and immunoblotting

Cells were lysed in lysis buffer (Cell Signaling Technology [CST], Danvers, MA, USA, #9803) supplemented with cOmplete tablet (Roche, Penzburg, Germany, #04693159001), phosSTOP (Roche #04906837001), and Benzonase nuclease (Sigma-Aldrich, St. Louis, MO, USA, E1014). Cell lysates were prepared by incubating with NuPage LDS Sample Buffer (Thermo Fisher, NP0007) and NuPAGE Sample Reducing Agent (Thermo Fisher, NP0004) for 5 min at 95°C to denature samples. Lysates were loaded onto NuPAGE 4–12% Bis-Tris gels (Thermo Fisher). Proteins were transferred to nitrocellulose membranes (Bio-Rad, Hercules, CA, USA) using Trans-Blot Turbo Transfer System (Bio-Rad). Membranes were blocked with Rockland blocking buffer at room temperature for 1 hr (Rockland Immunochemicals, Pottstown, PA, USA), incubated with primary antibody (diluted in Blocking Buffer) overnight at 4°C, and then with secondary antibodies (1:20,000 diluted in Blocking Buffer, LI-COR) for 1 hr at room temperature. Odyssey CLx Infrared Imaging System (LI-COR) was used for western blot detection and quantitation.

## Cell culture and treatment

HEK293T cells and A549 cells were cultured in DMEM media (Thermo Fisher #11965-092) containing 1% Pen/Strep and 10% FBS (VWR International, Radnor, PA, USA, #97068-085). For LLOMe treatment, LLOMe (Sigma-Aldrich, #L7393) was added at 1 mM for 2 hr or 4 hr prior to fixation or lysing cells for downstream analysis. Nocodazole (Sigma-Aldrich, #M1404) was added at 25 μM for 2 hr prior to fixation or live-cell imaging. Cells were routinely screened to confirm the absence of mycoplasma contamination.

## Generation of Dox-inducible cell lines expressing Rab12

Doxycycline-inducible cell lines were generated to stably express WT Rab12 or a phospho-deficient mutant of Rab12 (S106A) in RAB12 KO A549 cells. Briefly, lentiviral constructs were generated by cloning 3XFLAG-RAB12 (or RAB12 S106A) into the pLVX-TetOne-Puro vector. Lentivirus was produced by transfecting the plasmids in HEK293T cells using Lenti-X Packaging Single Shots (Takara Bio, Kusatsu, Shiga, Japan, #631278). The media containing lentivirus were collected from transfected cells and were further concentrated by 50-fold using Lenti-X Concentrator (Takara, #631231). RAB12 KO A549 cells were infected with lentivirus expressing WT 3XFLAG-RAB12 or 3XFLAG-RAB12 S106A mutant. Cells carrying the lentiviral vectors were selected with puromycin (1 μg/mL). To enable the expression of WT Rab12 or the Rab12 S106A mutant, doxycycline (0.1, 0.5, and 1 μg/mL) was added in the cell culture for 3 days.

## Immunoprecipitation of lysosomes using TMEM192-HA[x3]

Lysosomes were isolated by immunoprecipitation from cells expressing the TMEM192-HA[x3] transgene as described previously (*Abu-Remaileh et al., 2017*) with the following modifications. Cells were plated in 15 cm culture dishes such that they reached full confluency on the day of the experiment. All subsequent steps were performed at 4°C or on ice with pre-chilled reagents, unless otherwise noted. Media was removed and monolayers were rinsed with KPBS buffer (136 mM KCl, 10 mM $KH_2PO_4$, pH

7.25), harvested by scraping into fresh KPBS and pelleted via centrifugation. Cell pellets were resuspended in KPBS + buffer (KPBS supplemented with 3.6% [wt/vol] iodixanol [OptiPrep; Sigma-Aldrich], cOmplete protease inhibitor [Roche], and PhosStop phosphatase inhibitor [Roche]), and cells were fractionated by passing the suspension through a 21 G needle five times followed by centrifugation at 800 × g for 10 min. Post-nuclear supernatant (PNS) was harvested and incubated with anti-HA magnetic beads (pre-blocked with BSA and washed with KPBS buffer; Thermo Fisher) for 15 min with end-over-end rotation. Lysosome-bound beads were washed three times with KPBS + buffer, and samples used for immunoblotting were eluted from beads by heating to 95°C for 10 min in 1× NuPAGE LDS Sample Buffer (Thermo Fisher).

## Analysis of total and pRab10 and total and pRab12 levels on isolated lysosomes from WT, RAB12 KO, and PD-linked variant KI A549 cell models

For analysis of pRab10 levels on isolated lysosomes, one confluent 15 cm plate of cells was used per experimental condition. Cells were treated with 1 mM LLOMe (or vehicle) for 2 hr at 37°C prior to isolation of lysosomes via anti-HA immunoprecipitation as described above. Lyso-IP were performed with 60 µL of anti-HA magnetic bead slurry per condition. Immunoblotting for pRab10 and pRab12 levels was performed in parallel with analysis of total Rab10 and Rab12 levels, as detailed above, using 20% of total immunoprecipitated material per condition. pT73 Rab10, pS106 Rab12, total Rab10, total Rab12, and HA band intensities were quantified from immunoblots using ImageStudio Lite software (LI-COR), and the phospho- and total Rab band intensities were normalized to HA band intensity within each experimental condition. Data were normalized to the median value within each replicate and was then normalized to the mean value of vehicle-treated WT samples across replicates. Calculations for the total fraction of Rab12 present on immunoprecipitated lysosomes were performed by extrapolating the quantitated western blot signal of both the IP and PNS fractions out to 100%, and then calculating the percent of total estimated Rab12 signal captured in the IP divided by the total estimated Rab12 signal present in the PNS sample.

## Analysis of LRRK2 levels on isolated lysosomes from A549 cell models

For analysis of LRRK2 levels on isolated lysosomes, three confluent 15 cm plates of cells (seeded 24 hr prior to the assay start) were used per experimental condition. Cells were treated with 1 mM LLOMe (or vehicle) for 4 hr at 37°C and then lysosomes were isolated via anti-HA immunoprecipitation as described above. Lyso-IP were performed with 150 µL of anti-HA magnetic bead slurry per condition. For immunoblot detection of endogenous LRRK2, 25% of the total immunoprecipitated material (per condition) was loaded onto a 3–8% Tris-Acetate gel (Thermo Fisher), fully resolved gels were transferred to nitrocellulose membranes, probed overnight at 4°C with a 1/500 dilution of mouse anti-LRRK2 (clone N241A/34; UC Davis/NIH NeuroMab Facility, Davis, CA, USA), and imaged using standard immunoblotting protocol detailed above. LRRK2 and HA band intensity was quantified from immunoblots using ImageStudio Lite software (LI-COR), LRRK2 intensity was normalized to HA band intensity within each experimental condition, data was normalized to the median value within each replicate, and then was normalized to the mean value of vehicle-treated WT samples across replicates. Calculations for the total fraction of LRRK2 present on immunoprecipitated lysosomes were performed as for Rab12 (see above).

## Immunostaining of pT73 Rab10, Rab10, HA, and LAMP1 and image analysis

WT, RAB12 KO, and LRRK2 KO A549 cells were seeded in 96-well plates (Revvity, Waltham, MA, USA, Phenoplate, #6055302), and then treated with vehicle or LLOMe (1 mM). After 2 hr, cells were fixed with 4% PFA for 15 min, permeabilized and blocked with blocking buffer (5% Normal Donkey Serum/0.05% Triton X-100/PBS) for 1 hr at room temperature. Primary antibodies were diluted in blocking buffer and incubated overnight at 4°C. pT73 Rab10 antibody (Abcam, ab241060, 1:100), Rab10 antibody (CST, 8127, 1:100), LAMP1 antibody (Abcam, ab25630, 1:100), and HA antibody (CST, 3724, 1:100) were used in the study. After three washes with PBS/0.05% Triton X-100, secondary fluorescently labeled antibodies were diluted in blocking buffer and incubated for 1 hr at room temperature. DAPI (1:1000) and cell mask deep red (1:5000, Thermo Fisher, C10046) were diluted in PBS/0.05% Triton

X-100 and incubated for 10 min. After three washes with PBS/0.05% Triton X-100, the cell plates were imaged on an automated confocal high-content imaging system (Revvity, Opera Phenix Plus High-Content Screening System) using a 63× water immersion objective lens with excitation lasers (405 nm, 488 nm, 561 nm, 640 nm) and preset emission filters. Channels were separated to avoid fluorescence crosstalk. A custom analysis was developed in the Harmony 4.9 image analysis software (Revvity) to enable image analysis. For analysis of puncta intensity, pT73 Rab10 or total Rab10 spots were defined using 'Finding Spots' building blocks, and the sum of corrected spot intensity per cell was used to measure puncta signals. The colocalization between pT73 Rab10 or TMEM192-HA puncta and LAMP1-positive lysosomes were measured with object-based analysis. Briefly, pT73 Rab10 or TMEM192-HA and LAMP1 spots were independently defined using separate 'Find Spots' building blocks. Colocalized pT73 Rab10 and LAMP1 spots or TMEM192-HA and LAMP1 spots were identified using the geometric center overlap method within the 'Select Population' tool. The average number of colocalized spots were calculated per cell from 16 fields per well (for pT73 Rab10 and LAMP1 analysis) or 20 fields per well (for HA and LAMP1 analysis) and averaged across the well.

## Image analysis of the localization of Rab12 and LRRK2 to lysosomes and Golgi

For the colocalization analysis of Rab12 and organelle markers, HEK293T cells were transfected with mCherry-Rab12 plasmid (pcDNA3.1 vector) using Lipofectamine LTX with Plus Reagent (Thermo Fisher #15338100), and cells were plated onto poly-lysine-coated 96-well plates (Revvity, Waltham, MA, USA, Phenoplate, #6055302). Two days after transfection, cells were treated with vehicle or LLOMe (1 mM), with or without nocodazole (25 μM). After 2 hr, cells were fixed and immunostained with a LAMP1 antibody (Abcam, ab25630, 1:100) or GM130 antibody (Abcam ab52649). Cell plates were imaged on an automated confocal high-content imaging system (Revvity, Opera Phenix Plus High-Content Screening System) using a 40× water immersion objective lens. For the colocalization analysis of LRRK2 and organelle markers, HEK293T cells stably expressing eGFP-LRRK2 were used. For the colocalization analysis of LRRK2 and Rab12, HEK293T cells stably expressing eGFP-LRRK2 were transfected with mCherry-Rab12 plasmid. After LLOMe and nocodazole treatment, cell plates were imaged using a 63× water immersion objective lens.

For image analysis, we used PCC to assess the colocalization between various pairs of fluorophores: (1) mCherry-Rab12 and 488 LAMP1-positive lysosomes, (2) mCherry-Rab12 and 647 GM130-positive Golgi, (3) eGFP-LRRK2 and 568 LAMP1-positive lysosomes, (4) eGFP-LRRK2 and 647 GM130-positive Golgi, and (5) mCherry-Rab12 and eGFP-LRRK2, as the PCC is a commonly used intensity-based measurement to quantify colocalization between two fluorophores. A custom analysis algorithm built in the Harmony image analysis software (versions 5.1 and 4.9, Revvity) was used to calculate PCC. For Rab12 or LRRK2 singly expressing cells, individual nuclei were identified with DAPI and cell boundaries were defined using either the mCherry or GFP fluorescent channel, respectively. Rab12-positive and LRRK2-positive cells were selected after thresholding for extremely low or high intensities. Lysosomes and Golgi were independently segmented using the 'Find Spots' building block in Harmony, and PCC for either Rab12 or LRRK2 was calculated on a per-cell basis within these two compartments. The average PCC score was calculated from ~30 fields per well across three independent biological replicates.

For Rab12-LRRK2 colocalization, the number of Rab12-LRRK2 co-expressing cells was much lower than their single-expressing counterparts. Therefore, additional steps were taken to manually identify cells that expressed both mCherry-Rab12 and eGFP-LRRK2 at correct levels. mCherry-Rab12 and eGFP-LRRK2 PCC were calculated on a per-cell basis from ~3 wells across three independent biological replicates.

Calculation of the percentage of Rab12 and LRRK2 in lysosomes was performed using the Harmony image analysis software (versions 5.1 and 4.9, Revvity) to determine the percentage of mCherry-Rab12 intensity in lysosomes compared to the whole cell. From Rab12-positive cells, lysosomes were segmented using the LAMP1 channel. mCherry-Rab12 sum intensity from lysosomes was calculated and divided by the total sum mCherry-Rab12 intensity within the entire cell. Values were calculated on a per-cell basis, averaged across ~30 fields per well, and across three independent biological replicates. A similar image analysis process was adapted for calculating the percentage of LRRK2 within lysosomes.

## Image analysis of Rab12 in WT and LRRK2 KO cells

WT and LRRK2 KO A549 cells were transfected with mCherry Rab12 plasmid (pcDNA3.1 vector) using Lipofectamine LTX with Plus Reagent (Thermo Fisher #15338100), and cells were plated onto poly-lysine-coated 96-well plates (Revvity, Waltham, MA, USA, Phenoplate, #6055302). Two days after transfection, cells were treated with vehicle or LLOMe (1 mM). After 2 hr, cells were fixed and immunostained with LAMP1 antibody (Abcam, ab25630, 1:100). Cell plates were imaged on an automated confocal high-content imaging system (Revvity, Opera Phenix Plus High-Content Screening System) using a 63× water immersion objective lens.

For the image analysis, cells were identified through a nuclear stain (DAPI) and lysosomes were segmented with LAMP1 staining using the 'Find Image Region' building block in the Harmony 5.1 analysis software (Revvity). The total mCherry-Rab12 levels present in lysosomes were determined by calculating the mean fluorescence intensity of the 568 nm channel within the LAMP1 area. Values were measured from mCherry-Rab12 expressing cells (n=20 cells per condition, with cellular intensity between 2000 and 5000 fl. units) and averaged across wells (~4–6 wells per condition).

## Live-cell imaging of Rab12 and LRRK2 in HEK293T cells

HEK293T cells were transfected with eGFP-LRRK2 and mCherry-Rab12 plasmids (pcDNA3.1 vectors) using Lipofectamine LTX with Plus Reagent (Thermo Fisher #15338100), and cells were plated onto poly-lysine-coated 96-well plates (Corning Inc, Corning, NY, USA, BioCoat plates, #354640). Two days after transfection, cells were incubated with Hoechst 33342 (1 µg/mL, Thermo Fisher #62249) and CellMask Deep Red Plasma membrane Stain (1:2000, Thermo Fisher C10046) for 10 min. After replacing the cell culture media containing 1 mM LLOMe, the cell plates were immediately started for live-cell imaging on an automated spinning-disk confocal high-content imaging system (Revvity, Opera Phenix) using a 40× water immersion objective lens under 5% $CO_2$ and 37°C condition. The confocal images were taken every 10 min for 90 min in total.

## Time lapse cell segmentation analysis

Fields of view containing cells co-transfected for eGFP-LRRK2 and mCherry-Rab12 were manually selected from the time lapse dataset across three independent experiments. For each field, channel, and timepoint, the z-stack was converted to a 2D image by maximum intensity projection, then the background intensity was estimated by smoothing the image with a Gaussian filter with a kernel standard deviation of 50 pixels (~14.8 µm) using the ndimage module in scipy v1.9.3 (*Virtanen et al., 2020*). The background was subtracted from the original image and all pixels below the background intensity were set to zero. Background subtracted images were loaded into napari v0.4.17 (*Sofroniew et al., 2022*) as 2D+time images for segmentation.

To better visualize cells co-expressing low levels of LRRK2 and Rab12, the contrast limits were set to between 0 and 125 AU for LRRK2 and between 0 and 600 AU for Rab12. Cells were included in the segmentation if they (1) appeared morphologically healthy, (2) were visible throughout the time series, and (3) co-expressed LRRK2 and Rab12 at levels above background, but below the maximum contrast limit. Cells were segmented using the brush tool in napari using a brush size of 10 pixels (~3.0 µm). Cells were painted to the edge of the cell border including the nucleus but excluding signal from cell debris and other extracellular sources. Tracking cells across frames was typically possible through a combination of proximity and morphology, but where the assignment was ambiguous, those cells were excluded from further analysis (n=2). The resulting dataset contained 55 segmented cells.

For each cell, the non-background subtracted LRRK2 and Rab12 signals were extracted and normalized to between 0.0 and 1.0 by subtracting 200 AU and then dividing by 800 AU. Values above 1.0 or below 0.0 were set to 1.0 or 0.0, respectively. For all pixels under the cell mask, the PCC I was calculated between the normalized LRRK2 and Rab12 signals using the pearsonr function in the scipy.stats module. Cell properties such as area, perimeter, and mean intensity in each channel were extracted for each timepoint using the regionprops function in scikit-image v0.19.3 (*van der Walt et al., 2014*). Cells were filtered for quality by fitting a least squares line to the mean intensity of both LRRK2 and Rab12 signal for each and excluding any cells where the slopes were negative (n=31 cells excluded), resulting in 24 validated cell traces. Normalized intensity and correlation coefficients were plotted as mean of all 24 traces ± standard error (SEM) using Prism Version 9.5.1 (GraphPad).

## Statistical analysis

Data are shown as mean ± SEM, and all statistical analysis was performed in GraphPad Prism 9. Unpaired (or paired) t-tests were used for statistical analyses of experiments with two treatment groups. For more than two groups, analysis was performed using one-way analysis of variance (ANOVA) with Tukey's multiple comparison, one-way ANOVA with Sidak's multiple comparison test, one-way ANOVA with Dunnett's multiple comparison test, repeated measures one-way ANOVA with Dunnett's multiple comparison or two-way ANOVA with Sidak's test, as indicated in figure legends. Comparisons were considered significant where *, $p<0.05$; **, $p<0.01$; ***, $p<0.001$; ****, $p<0.0001$.

## Materials availability

The pT73 Rab10 antibody used in these studies is available from the corresponding author upon reasonable request.

## Acknowledgements

We thank members of the Denali post doc program and lysosomal function pathway team for useful discussions and feedback. We also thank Dario Alessi for his generosity in sharing his RAB29 KO A549 cell model.

---

## Additional information

### Competing interests

Xiang Wang, Oliver B Davis, Michael T Maloney, Maayan Agam, Marcus Y Chin, Rajarshi Ghosh, Dara E Leto, David Joy, Meredith EK Calvert, Joseph W Lewcock, Gilbert Di Paolo, Robert G Thorne, Anastasia G Henry: is an employee of Denali Therapeutics. Vitaliy V Bondar: was an employee of Denali Therapeutics when these studies were conducted and is currently an employee of REGENEXBIO Inc. Audrey Cheuk-Nga Ho: was an employee of Denali Therapeutics when these studies were conducted and is currently an employee of Cellares. Zachary K Sweeney: was an employee of Denali Therapeutics when these studies were conducted and is currently an employee of Interline Therapeutics Inc.

### Funding

No external funding was received for this work.

### Author contributions

Xiang Wang, Conceptualization, Formal analysis, Validation, Investigation, Visualization, Writing – original draft, Writing – review and editing; Vitaliy V Bondar, Conceptualization, Formal analysis, Investigation, Visualization, Methodology, Writing – review and editing; Oliver B Davis, Formal analysis, Validation, Investigation, Visualization, Writing – original draft, Writing – review and editing; Michael T Maloney, Conceptualization, Formal analysis, Validation, Investigation, Visualization, Methodology, Writing – original draft, Writing – review and editing; Maayan Agam, Validation, Investigation, Visualization; Marcus Y Chin, Formal analysis, Investigation, Visualization, Methodology, Writing – review and editing; Audrey Cheuk-Nga Ho, Formal analysis, Investigation, Visualization; Rajarshi Ghosh, Formal analysis, Investigation, Visualization, Methodology; Dara E Leto, Conceptualization, Methodology, Writing – review and editing; David Joy, Conceptualization, Formal analysis, Investigation, Visualization, Methodology, Writing – original draft; Meredith EK Calvert, Supervision, Writing – review and editing; Joseph W Lewcock, Robert G Thorne, Zachary K Sweeney, Conceptualization, Supervision, Writing – review and editing; Gilbert Di Paolo, Conceptualization, Writing – review and editing; Anastasia G Henry, Conceptualization, Formal analysis, Supervision, Visualization, Writing – original draft, Project administration, Writing – review and editing

### Author ORCIDs

Xiang Wang http://orcid.org/0000-0001-5134-1292
Oliver B Davis https://orcid.org/0000-0002-3622-8651
David Joy https://orcid.org/0000-0001-9941-9538

Joseph W Lewcock ⓘ https://orcid.org/0000-0003-3012-7881
Anastasia G Henry ⓘ https://orcid.org/0000-0002-8124-5477

**Decision letter and Author response**

Decision letter https://doi.org/10.7554/eLife.87255.sa1
Author response https://doi.org/10.7554/eLife.87255.sa2

## Additional files

### Supplementary files
• MDAR checklist

### Data availability
All data generated or analyzed during this study are included in the manuscript and supporting files; source data files for western blots have been provided for all figures.

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
