## [Editor Report]

This valuable study shows that Rab12 regulates LRRK2 activation via damaged lysosomes. The strength of evidence supporting the claim is compelling. Although key questions about the mechanism remain unanswered, these findings provide a useful template for further research.

---

## [Decision Letter]

**Decision letter after peer review:**

Thank you for submitting your article "Rab12 regulates LRRK2 activity by promoting its localization to lysosomes" for consideration by *eLife*. Your article has been reviewed by 3 peer reviewers, and the evaluation has been overseen by a Reviewing Editor and Vivek Malhotra as the Senior Editor. The reviewers have opted to remain anonymous.

Essential revisions:

1. Tone down conclusions throughout as indicated in the reviews.

2. Verify the LRRK2 dependence of activation of rab10 phosphorylation.

3. Demonstrate that Rab12 does in fact have a more important role than Rab29.

4. Clarify the discrepancy in Figure 4 (see comments).

5. The data do not show lysosomal accumulation of rab12 precedes LRRK2 (point #1 above) and clarify Fig. 3 as described by Reviewer #3.

*Reviewer #1 (Recommendations for the authors):*

1. Fig. 2G. The A549 R1441G cells show lower kinase activity than wild-type cells. Does this clone have less LRRK2? These cells should also be listed under Key Reagents.

2. Figure 3E would benefit from nocodazole addition to most clearly demonstrate the true co-localization of Rab12 and LRRK2 on lysosomes (rather than concentration in a very crowded perinuclear area).

3. Line 326. "Rab12 and LRRK2 showed a primarily cytosolic localization [at baseline]." Most Rabs are 50% on membranes or bound to GDI in cytosol. The images shown highlight massive concentration in one part of the cell upon LLOME addition and the faint, small-vesicle bound Rab12 would appear very diffuse without actually being non-membrane associated or "cytosolic". Perhaps the word "diffuse" would be more precise for the Rab?

4. In the Lyso-IPs, there seems to be a lot of Rab10 on lysosomes yet by microscopy that is never seen. Perhaps the tag is pulling down some Golgi membranes for biosynthetic trafficking of the HA-tag? It might be worth mentioning that this represents a very small fraction of total Rab10 if that is the case. Also, the LAMP is lost in LLOME samples and this should also be mentioned.

5. Normally mechanism is required for an *eLife* paper but this is a short report in a competitive area where other complementary stories will bolster this work (Biorxiv doi: https://doi.org/10.1101/2023.02.17.529028).

*Reviewer #3 (Recommendations for the authors):*

Although the overall role of Rab12 in activating LRRK2 is well supported by the data, there are several areas where the study could be improved.

1. Although it is likely that the reported relationship between Rab12 and LRRK2 is generalizable to other cell types, this remains to be shown outside of A549 cells. Unapanta et al, bioRxiv, 2022 showed that Rab38 can also play an important role in activating LRRK2 in melanocytes and other cell types might employ other LRRK2 activation mechanisms. Although it is beyond the scope of this study, it will eventually be important to see how Rab12 KO in a mouse model affects LRRK2 activity across a range of tissues (similar to what the Alessi lab did for Rab29).

2. It is puzzling that even though LLoMe-induced lysosome damage does not increase LRRK2-R1441G abundance at lysosomes (Fig 4D), it still promotes LRRK2-dependent phosphorylation of Rab10 by LRRK2-R1441G (Fig 4B). This result suggests that something more than recruitment to lysosomes is required for Rab12-dependent LRRK2 activation. At a minimum, some discussion of these results and the potential impacts of LRRK2 activation mechanisms would be helpful. The claim in lines 395-397 that these pathogenic variants are unable to further respond to additional lysosomal stress is undermined by the discrepancy between 4B vs 4D.

3. Based on data in Figure 4A, it is claimed that lysosome damage causes LRRK2 to accumulate at lysosomes. However, in Figure 4D the same treatment fails to yield a similar increase in LRRK2 levels at lysosomes. These inconsistencies undermine confidence in these results and should be explained.

4. In multiple figures, treatment with LLoMe results in a decrease in the amount of LAMP1 that is recovered in the TMEM192-HA "lysosome" immunoisolations. This effect was not explained. However, it raises a concern about whether LLoMe is truly triggering LRRK2 recruitment to lysosomes versus the possibility that there is also an effect on TMEM192-HA localization. Is LRRK2 getting recruited to lysosomes? Or is TMEM192-HA moving to a different compartment in the endolysosomal pathway? Some simple experiments that test the colocalization between TMEM192-HA and lysosome proteins +/- LLoMe treatment would help to address this.

5. Fig. 3A shows phosphorylated Rab12 at lysosomes but does not show total Rab12 at lysosomes. Showing total Rab12 is required to support claims of regulated recruitment.

6. The methods section lacks details for cell line generation and validation. How was Cas9 delivered? How were clones selected? How were mutations in Rab12 assessed?

7. Details are lacking on the source and identity of the TMEM192-3xHA plasmid.

[Editors’ note: further revisions were suggested prior to acceptance, as described below.]

Thank you for resubmitting your work entitled "Rab12 regulates LRRK2 activity by facilitating its localization to lysosomes" for further consideration by *eLife*. Your revised article has been evaluated by Vivek Malhotra (Senior Editor) and a Reviewing Editor.

The manuscript has been improved but there are some remaining issues that need to be addressed, as outlined below:

The main concerns that remained were:

1) Title is not appropriate. See below comments from Reviewer #1.

2) Some of the data interpretations were not appropriate. All reviewers commented on it.

3) Please discuss alternate models, see point 3 from Reviewer #2.

*Reviewer #1 (Recommendations for the authors):*

I was really hoping to move this forward; unfortunately, the response to the reviewer comments is not sufficient as follows.

1. The authors were asked to modify the title. As it stands, the title implies that Rab12 is on lysosomes which it is normally not, and that LRRK2 is only active on lysosomes which is also not true. The title is really important and either needs to indicate that Rab12 promotes localization to DAMAGED lysosomes or be otherwise modified.

2. LRRK2 still produces pRab10 in Rab12 knockout cells. The authors claim on Page 13 that Rab12 is necessary and sufficient for Rab10 phosphorylation but this is not correct. It is clear that LRRK2 still produces pRab10 in cells lacking Rab12 from blots in 2A and 1D. In Figure 2C, the authors show significant variability in Rab10 levels. This highlights the importance of normalizing phosphoRab10 levels to total Rab10 levels. Please normalize data to total Rab10 and not GAPDH or other protein. This is standard in this field.

3. The only data on Rab12 being on lysosomes comes from lyso-IPs. In order to understand the magnitude of the process being studied here the authors need to state the fraction of the total Rab12 and total LRRK2 in the LysoIPs of Fig. 3 and 4. This is essential. For example, it is possible that Rab12 increases LRRK2 on lysosomes from 2% of total to 4% of total. The reader needs this information.

4. The authors were asked to carry out nocodazole experiments to document true lysosome co-localization. 293T cells are poorly adherent and give poor microscopy images with or without LLOME or nocodazole. Why didn't the authors use their A549 cells as in Fig. 3E? Note that a Golgi marker in these cells would have shown complete co-localization with lysosomes in this scenario-this is why nocodazole is so important to separate membrane compartments. The authors need a clear and well controlled experiment showing that the structures they visualize are lysosomes and not Golgi derived vesicles. This circles back to the title of the paper.

*Reviewer #2 (Recommendations for the authors):*

The authors have responded to all of the concerns raised. It is nice to see that LRRK2 is not required for the localization of rab12 to the lysosome. What is very clear is that LLOMe recruits rab12 to the lysosome, induces phosphorylation of rab10 and rab12 is required for phosphorylation of rab10. However, a discrepancy remains between the recruitment of LRRK2 to the lysosome by LLOMe, which is not as apparent for wt LRRK2 in Fig. 4D as in 4A, and the phosphorylation of rab10 induced by LLOMe (4B,C). So it seems likely, as the authors have now included in the results, that mechanisms other than lysosomal recruitment of LRRK2 contribute to the phosphorylation of rab10. Importantly, they have also shown that LRRK2 is responsible for the induction by LLOMe (Fig. 2), so I think this is fine, but they should probably include this important consideration in the discussion as well as the results.

*Reviewer #3 (Recommendations for the authors):*

The revised manuscript provides strong evidence that Rab12 can regulate the ability of LRRK2 to phosphorylate Rab10 and that this occurs at least in part by regulating the abundance of LRRK2 at lysosomes. Although key questions about the mechanism remain unanswered, these are still important findings. New data was added to the manuscript that helped to address concerns that were raised during the previous review. Below I have provided some suggestions for clarification to the text that should be addressable without the need for new experiments. However, I also remain concerned about some important results and their interpretation (points 1-).

1. There is a discrepancy between the strong decrease in LAMP1 levels in the lysosome IP samples in Figure 3A and that is not seen in the LAMP1 immunofluorescence signal in 3C. This should be acknowledged and/or explained in the text.

2. Lines 272-275 imply that previous studies (references 32 and 33) also saw reductions in LAMP1 levels on purified lysosomes following LLoMe treatment. However, I could not find evidence of this in the papers that were cited.

3. There is still a discrepancy between Figure 4A-C where LLoMe activates LRRK2 (as measured by phosphor-Rab10) even in the context of VPS35-D620N and LRRK2-R1441G and Figure 4D where LLoMe does not change LRRK2 lysosome abundance in the same cells. The authors touched on this in the response to reviewers but the answer was not definitive. The description and interpretation of these results in the manuscript text is confusing as it is unclear to what extent LRRK2 kinase activity is regulated in response to these stimuli versus LRRK2 localization and proximity to substrates. The data suggests that there might be independent changes to LRRK2 localization and activity. It is not necessary for the authors to solve these puzzles but they should be more clearly presented.

4. Line 168: The statement that Rab12 is necessary and sufficient is too strong. Some commentary is required to at minimum acknowledge that this study only examined A549 cells and that the extent that the findings are generalizable to other cell types remains unknown.

5. Figure 3A and D and Figure 4A-D should also show the total cell lysates rather than just the IP results.

6. Figure 4E and Line 505 over-reach in stating that Rab12 mediates LRRK2 recruitment to lysosomes. Proving this would require evidence that a direct interaction between LRRK2 and Rab12 is necessary for LRRK2 recruitment and activation.

---

## [Author Response]

Essential revisions:Reviewer #1 (Recommendations for the authors):1. Fig. 2G. The A549 R1441G cells show lower kinase activity than wild-type cells. Does this clone have less LRRK2? These cells should also be listed under Key Reagents.

The LRRK2 R1441G KI A549 cells show reduced Rab10 phosphorylation when assessed at the whole cell level. This variant is also associated with reduced LRRK2 levels as assessed by MSD-based analysis in cell lysates as shown in Author response image 1, which likely contributes to the lower level of Rab10 phosphorylation observed with the LRRK2 R1441G variant in whole cell lysates.

**Author response image 1. sa2fig1:** 

However, analysis of both LRRK2 levels and Rab10 phosphorylation on lysosomes isolated from LRRK2 R1441G KI cells shows an increase in both the lysosomal levels of LRRK2 and pT73 Rab10 with this PD-linked variant. These data demonstrate that the LRRK2 R1441G variant leads to a relative increase in LRRK2 activity at the lysosome.We have updated the Key Reagents list to also include all knock-in and knockout A549 cell models that were employed in this study. We thank the reviewer for pointing this omission out in our initial submission.

2. Figure 3E would benefit from nocodazole addition to most clearly demonstrate the true co-localization of Rab12 and LRRK2 on lysosomes (rather than concentration in a very crowded perinuclear area).

As suggested by the reviewer, we employed nocodazole treatment to assess whether Rab12 and LRRK2 colocalization is maintained when the clustering of late endosomes/lysosomes in the perinuclear region is disrupted by blocking microtubule assembly. Briefly, HEK293 cells overexpressing mCherry-Rab12 and eGFP-LRRK2 were pre-treated with nocodazole (50 uM) for 2.5 hours and then treated with either vehicle or LLOMe (1mM) for 1.5 hours in the continued presence of nocodazole. The distribution of mCherry-Rab12 and eGFP-LRRK2 was then assessed using high-content confocal imaging. Unfortunately, treatment with both LLOMe and nocodazole caused significant cytotoxicity, rendering us unable to quantify the extent of colocalization of Rab12 and LRRK2 with and without nocodazole treatment following lysosomal membrane permeabilization. We were able to observe a few examples of surviving cells in our studies. In these cells, we found that the apparent colocalization of Rab12 and LRRK2 (overlay in white) following LLOMe treatment was maintained with nocodazole treatment (see example images in Author response image 2; scale bar: 20μm). These data suggest that the significant colocalization between Rab12 and LRRK2 observed in our manuscript is not simply due to increased concentration of vesicular structures near the perinuclear region. Because of the severe toxicity observed in our experiments, we prefer not to include this data in the revised manuscript.

3. Line 326. "Rab12 and LRRK2 showed a primarily cytosolic localization [at baseline]." Most Rabs are 50% on membranes or bound to GDI in cytosol. The images shown highlight massive concentration in one part of the cell upon LLOME addition and the faint, small-vesicle bound Rab12 would appear very diffuse without actually being non-membrane associated or "cytosolic". Perhaps the word "diffuse" would be more precise for the Rab?

As suggested by the reviewer, we have replaced “cytosolic” with “diffuse” to more precisely describe the localization of Rab12 observed in our imaging experiments.

4. In the Lyso-IPs, there seems to be a lot of Rab10 on lysosomes yet by microscopy that is never seen. Perhaps the tag is pulling down some Golgi membranes for biosynthetic trafficking of the HA-tag? It might be worth mentioning that this represents a very small fraction of total Rab10 if that is the case. Also, the LAMP is lost in LLOME samples and this should also be mentioned.

The reviewer raises valid points regarding the use of Lyso-IP method to isolate lysosomes following membrane damage. While we do observe significant enrichment of lysosomal proteins using the Lyso-IP method, we cannot rule out the possibility that some of the Rab10 detected arises from Golgi components that are also immunoprecipitated from biosynthetic trafficking of TMEM192-3xHA. To address this concern, we employed imaging-based methods to assess the colocalization of phospho-Rab10 with lysosomal markers at baseline and following LLOMe treatment in WT and RAB12 KO cells (Figure 3C). Using this orthogonal approach, we also observed significant colocalization of pT73 Rab10 with the lysosomal marker LAMP1 following LLOMe treatment that is abolished by RAB12 deletion or LRRK2 deletion. These results provide additional confirmation that LRRK2 promotes Rab10 phosphorylation on lysosomes in a Rab12-dependent manner. As suggested by the reviewer, we have added new text (lines 326-328 of the updated manuscript) to address this point: “While Rab10 has been reported to primarily localize to the Golgi and endosomes, our data show that a proportion of Rab10 is localized to lysosomes basally and in response to lysosomal damage(34, 35).

The reviewer also astutely noted the loss of LAMP1 detected in our isolated lysosomes following LLOMe treatment. This is an effect we have observed across our Lyso-IP experiments. Our data is consistent with previous studies that have also observed a loss of lysosomal transmembrane proteins following LLOMe treatment (Lee et al *Dev Cell* 2020 PMID: 32916093 and Eriksson et al *Cell Death Dis* 2020 PMID: 32409651). Our hypothesis is that the loss of LAMP1 following lysosomal membrane permeabilization may arise from the redistribution of LAMPs to the cytosol in small vesicles and/or targeted degradation via lysophagy, as suggested by the studies highlighted. Whereas LAMP1 levels are robustly reduced following LLOMe treatment, we observed a minimal impact on the levels of exogenously-expressed TMEM192-3xHA in isolated lysosomes (see Figure 3A and D and Figure 4A). As suggested by the reviewer, we have added new text to our updated manuscript that discusses the reduction in LAMP1 observed following LLOMe treatment and potential explanations for this effect (lines 299-304).

To increase confidence in the use of TMEM192-3xHA to isolate damaged lysosomes, we have also added new data to the updated manuscript assessing the localization of TMEM192-3xHA to lysosomes at baseline and following lysosomal membrane damage via LLOMe. We confirmed that LLOMe treatment did not impact the lysosomal localization of TMEM192-3xHA, and this new data has been added to Figure 3- figure 1 supplement and the modified text is on page 15 of the revised manuscript.

These data suggest that while LLOMe treatment results in lysosomal membrane damage, sufficient lysosomal integrity remains to enable purification of this subcellular compartment using TMEM192.

5. Normally mechanism is required for an eLife paper but this is a short report in a competitive area where other complementary stories will bolster this work (Biorxiv doi: https://doi.org/10.1101/2023.02.17.529028).

We agree with the reviewer that the observations from this study open up many interesting questions regarding the mechanism by which Rab12 senses lysosomal damage and promotes enhanced LRRK2-dependent phosphorylation of Rab10 on lysosomes. We have submitted this as a short report to communicate our findings in a timely fashion given the competitive nature of this field, and we hope to provide additional mechanistic insight regarding this process in future work.

Reviewer #3 (Recommendations for the authors):Although the overall role of Rab12 in activating LRRK2 is well supported by the data, there are several areas where the study could be improved.1. Although it is likely that the reported relationship between Rab12 and LRRK2 is generalizable to other cell types, this remains to be shown outside of A549 cells. Unapanta et al, bioRxiv, 2022 showed that Rab38 can also play an important role in activating LRRK2 in melanocytes and other cell types might employ other LRRK2 activation mechanisms. Although it is beyond the scope of this study, it will eventually be important to see how Rab12 KO in a mouse model affects LRRK2 activity across a range of tissues (similar to what the Alessi lab did for Rab29).

The reviewer raises a great question regarding whether Rab12 regulates LRRK2 activity in additional cell types beyond the A549 cell model used in the present work. In a recent preprint that also identified Rab12 as a key regulator of LRRK2 activation, the authors characterize the impact of loss of Rab12 *in vivo* by analyzing LRRK2-dependent Rab10 phosphorylation across different tissues in *Rab12* KO mice (Dhekne et al bioRxiv 2023 doi: https://doi.org/10.1101/2023.02.17.529028). The authors observed reductions in phospho-Rab10 levels in *Rab12* KO mouse lung with trends toward reduction in the large intestine and kidney. This study supports the relevance of Rab12 as a regulator of LRRK2 activity beyond the A549 cell model used in the present work. The authors could not assess the role for Rab12 in regulating LRRK2 activity in the brain given challenges in detecting pT73 Rab10 in brain tissue, and additional studies in human iPSC-derived CNS cells are warranted to better assess the consequences of Rab12 deletion in the brain. These additional studies will undoubtedly be a focus of future work by us (and others) but are beyond scope for the present manuscript.

2. It is puzzling that even though LLoMe-induced lysosome damage does not increase LRRK2-R1441G abundance at lysosomes (Fig 4D), it still promotes LRRK2-dependent phosphorylation of Rab10 by LRRK2-R1441G (Fig 4B). This result suggests that something more than recruitment to lysosomes is required for Rab12-dependent LRRK2 activation. At a minimum, some discussion of these results and the potential impacts of LRRK2 activation mechanisms would be helpful. The claim in lines 395-397 that these pathogenic variants are unable to further respond to additional lysosomal stress is undermined by the discrepancy between 4B vs 4D.

The reviewer raises an important point regarding the potential mechanisms by which the LRRK2 R1441G variant leads to increased Rab10 phosphorylation at the lysosome and whether additional factors beyond LRRK2 localization may contribute to this effect. We focused on detecting effects on the localization of endogenously-expressed LRRK2 to avoid any potential artifacts caused by LRRK2 overexpression. Because of this, the amount of LRRK2 detected on isolated lysosomes is low, and technical variability in lysosomal isolations across independent experiments can complicate our ability to quantify smaller effects on LRRK2 levels on lysosomes. To add confidence in the effect of lysosomal damage on LRRK2 levels in LRRK2 R1441G and VPS35 D620N cells, we performed two additional, independent replicates. The updated compiled data confirm that both variants have elevated levels of LRRK2 on isolated lysosomes at baseline compared to WT cells and show a non-significant trend of a further increase in LRRK2 levels following LLOMe treatment (updated Figure 4D).

These data suggest that increased LRRK2 recruitment to lysosomes upon damage may at least, in part, contribute to elevated Rab10 phosphorylation observed in LRRK2 R1441G and VPS35 D620N KI cells with LLOMe treatment. The extent of lysosomal Rab10 phosphorylation is far greater than the extent of increase in LRRK2 levels following LLOMe treatment, suggesting that additional mechanisms beyond increased recruitment of LRRK2 to lysosomes may contribute to this effect on LRRK2 activation as suggested by the reviewer.

In addition to updating Figure 4 with this new data, we have also modified the text to remove our conclusion that LLOMe treatment does not further impact LRRK2 levels in PD-linked variant cells and added new text to the results section (as described above) to highlight that additional mechanisms beyond increased LRRK2 recruitment to lysosomes may also contribute to LRRK2 activation following lysosomal damage in these cells.

3. Based on data in Figure 4A, it is claimed that lysosome damage causes LRRK2 to accumulate at lysosomes. However, in Figure 4D the same treatment fails to yield a similar increase in LRRK2 levels at lysosomes. These inconsistencies undermine confidence in these results and should be explained.

The reviewer astutely points out that Figure 4A shows a significant increase in LRRK2 levels on lysosomes in WT cells following LLOMe treatment, while Figure 4D shows a non-significant trend toward an increase in lysosomal LRRK2 levels under similar conditions. As described above, we think this variability is due to the low levels of LRRK2 detected on lysosomes upon endogenous expression and technical variability in recovery of lysosomes across experiments using the Lyso-IP method.

Our observations are consistent with previous work (Eguchi et al *PNAS* 2018 PMID: 30209220, Herbst et al *EMBO J* 2020 PMID: 32643832, and Bonet-Ponce et al *Science Adv* 2020 PMID: 33177079), and further supported by orthogonal imaging-based studies (Figure 3F-I), increasing confidence that lysosomal damage causes LRRK2 to accumulate at lysosomes.

4. In multiple figures, treatment with LLoMe results in a decrease in the amount of LAMP1 that is recovered in the TMEM192-HA "lysosome" immunoisolations. This effect was not explained. However, it raises a concern about whether LLoMe is truly triggering LRRK2 recruitment to lysosomes versus the possibility that there is also an effect on TMEM192-HA localization. Is LRRK2 getting recruited to lysosomes? Or is TMEM192-HA moving to a different compartment in the endolysosomal pathway? Some simple experiments that test the colocalization between TMEM192-HA and lysosome proteins +/- LLoMe treatment would help to address this.

The reviewer raises a valid concern regarding whether LLOMe treatment might confound our ability to successfully isolate lysosomes following lysosomal membrane permeabilization. As described in more detail in our response to point #4 from Reviewer #1, we consistently observe a depletion in LAMP1 levels on isolated lysosomes following LLOMe treatment while the lysosomal levels of TMEM192-3xHA were minimally impacted. Our hypothesis is that the loss of LAMP1 following lysosomal membrane permeabilization may arise from the redistribution of LAMPs to the cytosol in small vesicles and/or targeted degradation via lysophagy in response to lysosomal damage, as suggested by Lee et al *Dev Cell* 2020 PMID: 32916093 and Eriksson et al *Cell Death Dis* 2020 PMID: 32409651. Based on the reviewer’s feedback, we have added new text to our updated manuscript that discusses the reduction in LAMP1 observed following LLOMe treatment and potential explanations for this effect (lines 299-304).

We thank the reviewer for their suggestion to examine the lysosomal localization of TMEM192-3xHA with and without LLOMe treatment to increase confidence in the use of Lyso-IP to isolated damaged lysosomes. We assessed the colocalization of TMEM192-HA and the lysosomal marker LAMP1 treated with vehicle or LLOMe using high-content confocal imaging. We quantified the extent of colocalization between HA-tagged TMEM192 and LAMP1 and confirmed that LLOMe treatment has no impact on the localization of TMEM192 to lysosomes. This new data add confidence in the use of the Lyso-IP method to isolate damaged lysosomes and has been added to Figure 3- figure supplement 1.

5. Fig. 3A shows phosphorylated Rab12 at lysosomes but does not show total Rab12 at lysosomes. Showing total Rab12 is required to support claims of regulated recruitment.

We agree with the reviewer that it is important to show total Rab12 levels on isolated lysosomes at baseline and following LLOMe treatment to support our claims of regulated recruitment. This data can be found in Figure 3D of our updated manuscript, showing that Rab12 levels are significantly increased on lysosomes following LLOMe treatment and that this signal is lost in RAB12 KO cells to confirm the specificity of our antibody.

6. The methods section lacks details for cell line generation and validation. How was Cas9 delivered? How were clones selected? How were mutations in Rab12 assessed?

We thank the reviewer for suggesting that we expand on our methods section with f urther details on how our cell lines were generated and validated. Ribonucleoproteins containing the Cas9 protein and synthetic chemically modified sgRNA produced at Synthego were electroporated into the cells using Synthego's optimized protocol. Editing efficiency was assessed upon recovery, 48 hours post electroporation. Specifically, genomic DNA was extracted from a portion of the cells, PCR amplified and sequenced using Sanger sequencing. The resulting chromatograms are processed using Synthego Inference of CRISPR edits software (ice.synthego.com). To isolate monoclonal cell populations, edited cell pools were seeded at 1 cell/well using a single cell printer into 96 or 384 well plates. All wells were imaged every 3 days to ensure expansion from a single-cell clone. Clonal populations were screened and validated using the PCR-Sanger-ICE genotyping strategy to confirm successful gene deletion or editing.

We have updated our Materials and Methods section to describe how these cell lines were generated and validated.

7. Details are lacking on the source and identity of the TMEM192-3xHA plasmid.

We have added additional details on the generation of the TMEM192-3xHA plasmid and the construct itself to the Materials and Methods section.

[Editors’ note: further revisions were suggested prior to acceptance, as described below.]

Reviewer #1 (Recommendations for the authors):I was really hoping to move this forward; unfortunately, the response to the reviewer comments is not sufficient as follows.1. The authors were asked to modify the title. As it stands, the title implies that Rab12 is on lysosomes which it is normally not, and that LRRK2 is only active on lysosomes which is also not true. The title is really important and either needs to indicate that Rab12 promotes localization to DAMAGED lysosomes or be otherwise modified.

We agree with the reviewer that the title for our manuscript is very important and something we want to make sure best describes the findings from our study. Based on the reviewer’s comment, we had modified our title in our last resubmission to “Rab12 regulates LRRK2 activity by facilitating its localization to lysosomes” to soften our language around the requirement of Rab12 in mediating LRRK2’s localization to the lysosomes. We now appreciate that this did not fully address the reviewer’s concern around the role that lysosomal damage plays in this recruitment. Based on the reviewer’s feedback, we have further modified our title to “Rab12 is a regulator of LRRK2 activity and mediates its localization to damaged lysosomes.” We thank the reviewer for their additional feedback on the title and are happy to further adjust the title as needed.

2. LRRK2 still produces pRab10 in Rab12 knockout cells. The authors claim on Page 13 that Rab12 is necessary and sufficient for Rab10 phosphorylation but this is not correct. It is clear that LRRK2 still produces pRab10 in cells lacking Rab12 from blots in 2A and 1D. In Figure 2C, the authors show significant variability in Rab10 levels. This highlights the importance of normalizing phosphoRab10 levels to total Rab10 levels. Please normalize data to total Rab10 and not GAPDH or other protein. This is standard in this field.

We appreciate the reviewer’s comments around the requirement of Rab12 for LRRK2-dependent phosphorylation of Rab10, and we have provided new data and further revised the text to address these points. Using our sensitive MSD-based assay to quantify pT73 Rab10 levels under basal conditions, we observed a similar reduction in Rab10 phosphorylation with Rab12 knockdown or deletion as we also observed with LRRK2 knockdown or deletion (Figure 1A, Figure 2A and B) which led us to make this comment. As the reviewer noted, we do observe some residual Rab10 phosphorylation upon Rab12 knockdown/deletion when assessed by western blot analysis (Figure 2A and Figure 1D). This signal is also observed in LRRK2 KO A549 cells, which may be a result of background signal from western blot analysis or may also suggest that a minor proportion of Rab10 phosphorylation may be LRRK2- and Rab12- independent in this cell model as proposed by the reviewer. We appreciate this reviewer’s point and have therefore modified the text to remove any reference to Rab12 being required or sufficient for LRRK2-dependent Rab10 phosphorylation. Specifically, we have changed the subsection heading of the results section on page 13 from “Rab12 is necessary and sufficient for LRRK2-dependent phosphorylation of Rab10" to “Rab12 deletion attenuates LRRK2-dependent phosphorylation of Rab10.”

The reviewer also commented on variability in total Rab10 levels across clones examined in Figure 2C and suggested we normalize pRab10 levels to total Rab10 levels. Phospho-Rab10 levels in Figure 2B were measured using a quantitative MSD-based assay and normalized for protein input. Issues with specificity and sensitivity precluded us from using an MSD-based assay against total Rab10, so western blot analysis was used to measure total Rab10 levels. To address the reviewer’s concern, we have now normalized pRab10 levels (quantified using an MSD-based assay) to total Rab10 levels measured using western blot analysis. These new data have been added as a new supplementary figure (Figure 2- figure supplement 1B and G). These new data confirm our original findings in Figure 2B, strongly suggesting that alterations in Rab10 levels do not explain the significant reduction in pRab10 levels observed in Rab12 KO cells. They are presented on page 13 of the revised manuscript.

3. The only data on Rab12 being on lysosomes comes from lyso-IPs. In order to understand the magnitude of the process being studied here the authors need to state the fraction of the total Rab12 and total LRRK2 in the LysoIPs of Fig. 3 and 4. This is essential. For example, it is possible that Rab12 increases LRRK2 on lysosomes from 2% of total to 4% of total. The reader needs this information.

We thank the reviewer for these comments and have now added several additional data panels related to them that we feel have strengthened the paper. To address the reviewer’s question regarding the percentage of total LRRK2 and Rab12 present on isolated lysosomes, we quantified LRRK2, Rab12, and HA band intensity in WT cells from both the IP and PNS (post-nuclear supernatant, the input fraction used for the IP) samples used for Figure 3D and 4A. We then calculated the total amount of each species present in both sample types by extrapolation based on the amount of sample loaded for western blot analysis and determined the total percentage of each species recovered in the IP. This analysis indicates that ~2% of LRRK2 is on lysosomes at baseline, and LLOMe treatment roughly doubles this to ~4% present on lysosomes. Similarly, ~1% of total Rab12 is present on lysosomes at baseline, and LLOMe treatment increased this to ~1.5% present on lysosomes. This analysis has been added as new data to our updated manuscript in Figure 3- figure supplement 2G and is presented on page 22 of the revised manuscript. Our analysis of total HA recovery in the IP samples confirms that less than 100% of the total HA-labeled lysosomes present in the input were captured in all cases – by experimental design to ensure equal capture across conditions– which ultimately suggests that the calculated total LRRK2 and Rab12 levels present on lysosomes are likely underestimates of the true values.

We had previously included imaging data that showed Rab12 localization to lysosomes basally that was significantly increased upon LLOMe treatment (Figure 3E). As an orthogonal approach to address the reviewer’s question, we quantified the percentage of LRRK2 and Rab12 on lysosomes at baseline and following LLOMe treatment using imaging-based analysis. Specifically, we overexpressed either fluorescently-tagged LRRK2 or Rab12 and quantified the % of LRRK2 or Rab12 on lysosomes by measuring the amount of LRRK2 or Rab12 signal that colocalized with LAMP1 normalized to the total LRRK2 or Rab12 signal per cell. Our imaging-based analysis suggested that ~5% of LRRK2 localized to lysosomes at baseline and that LLOMe significantly increased this to ~6%, and ~12% of Rab12 was observed at lysosomes basally and ~14% following LLOMe treatment. These data have been added as a new figure panel in Figure 3- figure supplement 2H and is presented on page 22 of the revised manuscript. This analysis required overexpression of both LRRK2 and Rab12 which also has its caveats with respect to recapitulating endogenous localization but nonetheless provides a useful complimentary approach.

We employed two distinct methods to estimate the percentage of LRRK2 and Rab12 on lysosomes basally and following LLOMe treatment. Both methods support our conclusions that a portion of total LRRK2 and Rab12 localize to lysosomes and that lysosomal damage further increases their lysosomal localization. The potential caveats associated with both approaches make it difficult in the end for us to be confident about the precise quantification of both proteins on lysosomes. Nevertheless, we do agree with the reviewer that this data addresses an essential question and have therefore decided to include it with the caveat as stated on lines 733-738:

“These results also show that a small percentage of Rab12 and LRRK2 are present on lysosomes at baseline and that lysosomal damage leads to a significant increase in the localization of both proteins to the lysosome (Figure 3- figure supplement 2), but precise quantification of the amount of Rab12 and LRRK2 on lysosomes under these conditions is difficult and warrants further study.”

4. The authors were asked to carry out nocodazole experiments to document true lysosome co-localization. 293T cells are poorly adherent and give poor microscopy images with or without LLOME or nocodazole. Why didn't the authors use their A549 cells as in Fig. 3E? Note that a Golgi marker in these cells would have shown complete co-localization with lysosomes in this scenario-this is why nocodazole is so important to separate membrane compartments. The authors need a clear and well controlled experiment showing that the structures they visualize are lysosomes and not Golgi derived vesicles. This circles back to the title of the paper.

We thank the reviewer for their suggestion to use nocodazole to confirm the colocalization of LRRK2 and Rab12 and to more clearly understand if LRRK2 and Rab12 are recruited to lysosomes rather than the Golgi upon LLOMe treatment. In line with their suggestions, we have conducted new studies and added new figures to the manuscript that we feel have further strengthened the manuscript.

To examine this, we conducted new studies using overexpression of eGFP-tagged LRRK2 and mCherry-tagged Rab12 to follow the localization of each protein at baseline and following lysosomal damage. We had initially tried to use A549 cells, but it turned out to be very challenging to identify a sufficient number of cells expressing both eGFP-LRRK2 and mCherry-Rab12 due to poor transfection efficiency in A549 cells (especially for eGFP-LRRK2 due to its large size). We therefore used HEK293T cells to address the reviewer’s comment. We further optimized the concentration and duration of nocodazole treatment based on previously-published work (Fasimoye et al PNAS 2023 and Berndsen et al *eLife*, 2019) and found that reducing the nocodazole treatment to 25 uM for 2 hours reduced toxicity and enabled subsequent analysis. We saw that the Golgi marker GM130 showed fragmented and dispersed punctate structures, confirming the effects of nocodazole treatment in this paradigm.

By performing this new imaging analysis, our data showed that:

1) LLOMe treatment significantly increased the colocalization of Rab12 with the lysosomal marker LAMP1 but not with the Golgi marker GM130 (as quantified by Pearson’s correlation coefficient). This co-localization was retained upon nocodazole treatment (see new data included in figure 3D and Figure 3-figure supplement 1). These data support our hypothesis that lysosomal damage increases the localization of Rab12 to lysosomes and not Golgi-derived vesicles.

2) LLOMe treatment also significantly increased the colocalization of LRRK2 with LAMP1 but not with the Golgi marker GM130 (quantified by Pearson’s correlation coefficient). As was observed with Rab12, this co-localization was retained upon nocodazole treatment (see new data in Figure 3G and Figure 3-figure supplement 2). These data add confidence to our conclusions that lysosomal damage increases the localization of LRRK2 to lysosomes.

3) To add confidence that Rab12 and LRRK2 colocalize following LLOMe treatment, we quantified their colocalization using Pearson’s correlation coefficient. We observed colocalization between Rab12 and LRRK2 following lysosomal damage and saw that this co-localization was preserved upon nocodazole treatment (new figure 3F). These data further support our conclusion that Rab12 and LRRK2 colocalize with one another following LLOMe treatment.

Overall, these data add confidence that a portion of LRRK2 and Rab12 localize to lysosomes a baseline, LLOMe treatment induces increased localization of both LRRK2 and Rab12 to lysosomes but not to the Golgi, and LRRK2 and Rab12 colocalize following lysosomal damage. We thank the reviewer for their suggestion that has further strengthened our conclusions.

Reviewer #2 (Recommendations for the authors):The authors have responded to all of the concerns raised. It is nice to see that LRRK2 is not required for the localization of rab12 to the lysosome. What is very clear is that LLOMe recruits rab12 to the lysosome, induces phosphorylation of rab10 and rab12 is required for phosphorylation of rab10. However, a discrepancy remains between the recruitment of LRRK2 to the lysosome by LLOMe, which is not as apparent for wt LRRK2 in Fig. 4D as in 4A, and the phosphorylation of rab10 induced by LLOMe (4B,C). So it seems likely, as the authors have now included in the results, that mechanisms other than lysosomal recruitment of LRRK2 contribute to the phosphorylation of rab10. Importantly, they have also shown that LRRK2 is responsible for the induction by LLOMe (Fig. 2), so I think this is fine, but they should probably include this important consideration in the discussion as well as the results.

We are pleased that our previous submission addressed all of the concerns raised by this reviewer and thank them for their suggestions that have substantially improved our manuscript. We agree that additional mechanisms beyond lysosomal recruitment of LRRK2 may contribute to the phosphorylation of Rab10. In addition to the text that was added in the results section around this in our last submission, we have updated the discussion to also include this important consideration as suggested by the reviewer. Specifically, we have added the following text to the Discussion (lines 908-915):

“We cannot rule out that additional mechanisms beyond increased proximity between LRRK2 and its Rab substrates may contribute to Rab10 phosphorylation following lysosomal damage, as the magnitude of Rab10 phosphorylation induced by LLOMe treatment was greater than the extent of LRRK2 recruitment to damaged lysosomes. Additional studies are warranted to determine how lysosomal membrane rupture triggers Rab12 recruitment, to identify other regulatory processes that may contribute to Rab phosphorylation upon lysosomal damage, and to better define how broadly such mechanisms are employed to drive LRRK2 activation in PD.”

Reviewer #3 (Recommendations for the authors):The revised manuscript provides strong evidence that Rab12 can regulate the ability of LRRK2 to phosphorylate Rab10 and that this occurs at least in part by regulating the abundance of LRRK2 at lysosomes. Although key questions about the mechanism remain unanswered, these are still important findings. New data was added to the manuscript that helped to address concerns that were raised during the previous review. Below I have provided some suggestions for clarification to the text that should be addressable without the need for new experiments. However, I also remain concerned about some important results and their interpretation (points 1-).1. There is a discrepancy between the strong decrease in LAMP1 levels in the lysosome IP samples in Figure 3A and that is not seen in the LAMP1 immunofluorescence signal in 3C. This should be acknowledged and/or explained in the text.

The reviewer correctly notes that we did not observe a significant decrease in LAMP1 signal assessed via imaging following LLOMe treatment, in contrast to the decrease observed in isolated lysosomes. LLOMe treatment leads to permeabilization of the lysosomal membrane, and these data may suggest that we are immunopurifying fragments of lysosomal membrane from which LAMP1 has been dissociated or perhaps degraded. We observed strong recruitment of Gal3 to isolated lysosomes following lysosomal damage, supporting some membrane integrity and glycocalyx remain after LLOMe treatment. Our imaging-based analysis did not show obvious differences in LAMP1 signal with LLOMe treatment, likely because our fixation and imaging conditions would capture lysosomal membrane fragments and dissociated LAMP1.

To acknowledge this point, we have added the following text to the results section (lines 287-295):

“While LLOMe treatment reduced the levels of lysosomal-associated membrane protein 1 (LAMP1) in isolated lysosomes, the levels and localization of TMEM192-3xHA, the lysosomal membrane protein used to isolate lysosomes, were not significantly impacted by LLOMe treatment (Figure 3A and Figure 3- figure supplement 1). We did not observe a loss of LAMP1 signal by immunofluorescence analysis, suggesting that LAMP1 may dissociate or be degraded from ruptured lysosomal membranes upon immunopurification. These data suggest that while LLOMe treatment results in lysosomal membrane damage, sufficient lysosomal integrity remains to enable purification of this subcellular compartment using TMEM192.”

2. Lines 272-275 imply that previous studies (references 32 and 33) also saw reductions in LAMP1 levels on purified lysosomes following LLoMe treatment. However, I could not find evidence of this in the papers that were cited.

We appreciate that the inclusion of references 32 and 33 in our previous draft following our comment around potential explanations for the reduction in LAMP1 levels observed on isolated damaged lysosomes was not clearly elaborated. References 32 and 33 had been previously included to support our hypothesis that LAMP1 levels may be reduced due to enhanced lysophagy or redistribution to small vesicles upon damage. These papers showed that LLOMe treatment led to the redistribution of LAMP2 to the cytoplasm or increased the turnover of LAMP1, respectively. However, based on the great point raised by this reviewer that LAMP1 signal loss was not observed in our imaging-based studies, our speculation is that we are immunopurifying lysosomal membrane fragments from which LAMP1 has dissociated or been degraded. We have revised the text as described above in Point 1 to better reflect our thinking on this and removed reference to these papers entirely in the updated manuscrpt.

3. There is still a discrepancy between Figure 4A-C where LLoMe activates LRRK2 (as measured by phosphor-Rab10) even in the context of VPS35-D620N and LRRK2-R1441G and Figure 4D where LLoMe does not change LRRK2 lysosome abundance in the same cells. The authors touched on this in the response to reviewers but the answer was not definitive. The description and interpretation of these results in the manuscript text is confusing as it is unclear to what extent LRRK2 kinase activity is regulated in response to these stimuli versus LRRK2 localization and proximity to substrates. The data suggests that there might be independent changes to LRRK2 localization and activity. It is not necessary for the authors to solve these puzzles but they should be more clearly presented.

We sincerely appreciate the reviewer’s feedback that the description and interpretation of our results around the magnitude of increase in Rab10 phosphorylation being greater than the increase in LRRK2 localization to lysosomes observed following LLOMe treatment should be edited to more clearly represent our ideas on the balance between activity versus localization. To further clarify our results and interpretation of our data, we have modified our results section to the following (lines 849-857):

“The levels of LRRK2 on lysosomes were significantly increased in untreated LRRK2 R1441G KI cells and VPS35 D620N KI cells, suggesting that enhanced localization of LRRK2 to lysosomes and proximity to its Rab substrates may contribute to the elevated Rab10 phosphorylation observed on lysosomes at baseline in these cells (Figure 4D). Rab10 phosphorylation was increased on lysosomes in response to LLOMe treatment while the levels of LRRK2 were not significantly impacted on lysosomes isolated from LRRK2 R1441G and VPS35 D620N KI cells, suggesting additional mechanisms beyond LRRK2 localization may also contribute to LRRK2 activation in response to lysosomal damage in these cells.”

Further, we have added the following text to the Discussion (lines 908-915):

“We cannot rule out that additional mechanisms beyond increased proximity between LRRK2 and its Rab substrates may contribute to Rab10 phosphorylation following lysosomal damage, as the magnitude of Rab10 phosphorylation induced by LLOMe treatment was greater than the extent of LRRK2 recruitment to damaged lysosomes. Additional studies are warranted to determine how lysosomal membrane rupture triggers Rab12 recruitment, to identify other regulatory processes that may contribute to Rab phosphorylation upon lysosomal damage, and to better define how broadly such mechanisms are employed to drive LRRK2 activation in PD.”

4. Line 168: The statement that Rab12 is necessary and sufficient is too strong. Some commentary is required to at minimum acknowledge that this study only examined A549 cells and that the extent that the findings are generalizable to other cell types remains unknown.

Based on the reviewer’s suggestions, we have removed our statement regarding Rab12 being necessary and sufficient. Specifically, we have revised the text in the results section (line 165 on page 10) from “Rab12 is necessary and sufficient for LRRK2-dependent phosphorylation of Rab10” to “Rab12 deletion attenuates LRRK2-dependent phosphorylation of Rab10.” Further, we have added the following text to the Discussion (lines 915-918 on page 31):

“As our work focused on the role of Rab12 in A549 cells, it will also be important to understand whether Rab12 similarly regulates LRRK2 activation and localization to damaged lysosomes in other cell types.”

5. Figure 3A and D and Figure 4A-D should also show the total cell lysates rather than just the IP results.

We thank the reviewer for their suggestion to include the results from total cell lysates in addition to our IP results. We have included side by side blots showing the western blot data from the immunoprecipitated samples as well as the PNS (post-nuclear supernatant, the input fraction used for the Lyso-IPs). These have been added to Figure 3- figure supplement 1B and F and Figure 4-figure supplement 1A-D.

6. Figure 4E and Line 505 over-reach in stating that Rab12 mediates LRRK2 recruitment to lysosomes. Proving this would require evidence that a direct interaction between LRRK2 and Rab12 is necessary for LRRK2 recruitment and activation.

We appreciate the reviewer’s point regarding softening our language around Rab12 mediating LRRK2 recruitment to lysosomes without direct evidence that an interaction between these proteins is required for LRRK2 recruitment to damaged lysosomes. We have revised Figure 4E from “Rab12 mediates LRRK2 recruitment and Rab phosphorylation on lysosomes” to “Rab12 regulates LRRK2 recruitment and Rab phosphorylation on lysosomes.” Further, we have revised the sentence in the discussion to remove “mediates” and instead use the word “regulates”:

“Our findings provide key insight into the mechanism by which LRRK2 activity is increased in response to lysosomal damage by demonstrating that Rab12 regulates LRRK2 localization to ruptured lysosomes.”